# Violence in fishing, hunting, and gathering societies of the Atacama Desert coast: A long-term perspective (10,000 BP—AD 1450)

Vivien G. Standen[1]*, Calogero M. Santoro[2], Daniela Valenzuela[1], Bernardo Arriaza[2], John Verano[3], Susana Monsalve[1], Drew Coleman[4], Pablo A. Marquet[5,¤a,¤b,¤c]

1 Departamento de Antropología, Universidad de Tarapacá, Arica, Chile, 2 Instituto de Alta Investigación, Universidad de Tarapacá, Arica, Chile, 3 Department of Anthropology, Tulane University, New Orleans, Los Angeles, United States of America, 4 Department of Geological Sciences, University of North Carolina, Chapel Hill, North Carolina, United States of America, 5 Departamento de Ecología, Facultad de Ciencias Biológicas, Pontificia Universidad Católica de Chile, Santiago, Chile

☉ These authors contributed equally to this work.
¤a Current address: Instituto de Ecología y Biodiversidad (IEB), Santiago, Chile
¤b Current address: Laboratorio Internacional en Cambio Global y Centro de Cambio Global UC, Pontificia Universidad Católica de Chile, Santiago, Chile
¤c Current address: The Santa Fe Institute, Santa Fe, New Mexico, United States of America
* vivien.standen@gmail.com

**Data Availability Statement:** All relevant data are within the manuscript and its Supporting information files.

## Abstract

In this study, we examine the long-term trajectory of violence in societies that inhabited the coast of the Atacama Desert in northern Chile using three lines of evidence: bioarchaeology, geoarchaeology and socio-cultural contexts (rock art, weapons, and settlement patterns). These millennia-old populations adopted a way of life, which they maintained for 10,000 years, based on fishing, hunting, and maritime gathering, complementing this with terrestrial resources. We analyzed 288 adult individuals to search for traumas resulting from interpersonal violence and used strontium isotopes $^{87}Sr/^{86}Sr$ as a proxy to evaluate whether individuals that showed traces of violence were members of local or non-local groups. Moreover, we evaluated settlement patterns, rock art, and weapons. The results show that the violence was invariant during the 10,000 years in which these groups lived without contact with the western world. During the Formative Period (1000 BC-AD 500), however, the type of violence changed, with a substantial increase in lethality. Finally, during the Late Intermediate Period (AD 1000–1450), violence and lethality remained similar to that of the Formative Period. The chemical signal of Sr shows a low frequency of individuals who were coastal outsiders, suggesting that violence occurred between local groups. Moreover, the presence of weapons and rock art depicting scenes of combat supports the notion that these groups engaged in violence. By contrast, the settlement pattern shows no defensive features. We consider that the absence of centralized political systems could have been a causal factor in explaining violence, together with the fact that these populations were organized in small-scale grouping. Another factor may have been competition for the same resources in the extreme environments of the Atacama Desert. Finally, from the Formative Period onward,

**Funding:** Grant: FONDECYT N°1171708 "Violencia en la prehistoria del Norte de Chile: un estudio multidisciplinario." The funders had no role in study design, data collection and analysis, decision to publish, or preparation of the manuscript.

**Competing interests:** The authors have declared that no competing interests exist.

we cannot rule out a certain level of conflict between fishers and their close neighbors, the horticulturalists.

## Introduction

We evaluate the long-term trajectory of violence in societies that inhabited the coast of the Atacama Desert in the extreme north of Chile (ca. 18°28´S 70°19´W) (Fig 1). These millennia-old populations had a way of life based on fishing, hunting, and maritime gathering, which they complemented with terrestrial resources. This traditional lifestyle was maintained for 10,000 years and continued for at least two centuries after contact with the first Europeans who arrived in this territory around AD 1536 [1]. This broad timescale gives us a unique opportunity to evaluate violence in a particular territory, and in populations that maintained a subsistence economy based essentially on the exploitation of fishing resources, later giving way to increasing social complexity and the emergence of chiefdoms.

In recent decades, archaeology, bioarchaeology, ethnohistory, ethnography, and primatology, employing both theoretical and empirical approaches, have brought an ever- increasing number of findings and robust data, suggesting that interpersonal violence and warfare would have played a role in the daily lives of many hunter-gatherer groups around the planet, from the late Pleistocene to modern times.

Several theoretical perspectives based on materialistic, ecological, evolutive and bioarcheological background compete to explain violence among hunter-gatherer societies [2]. The first

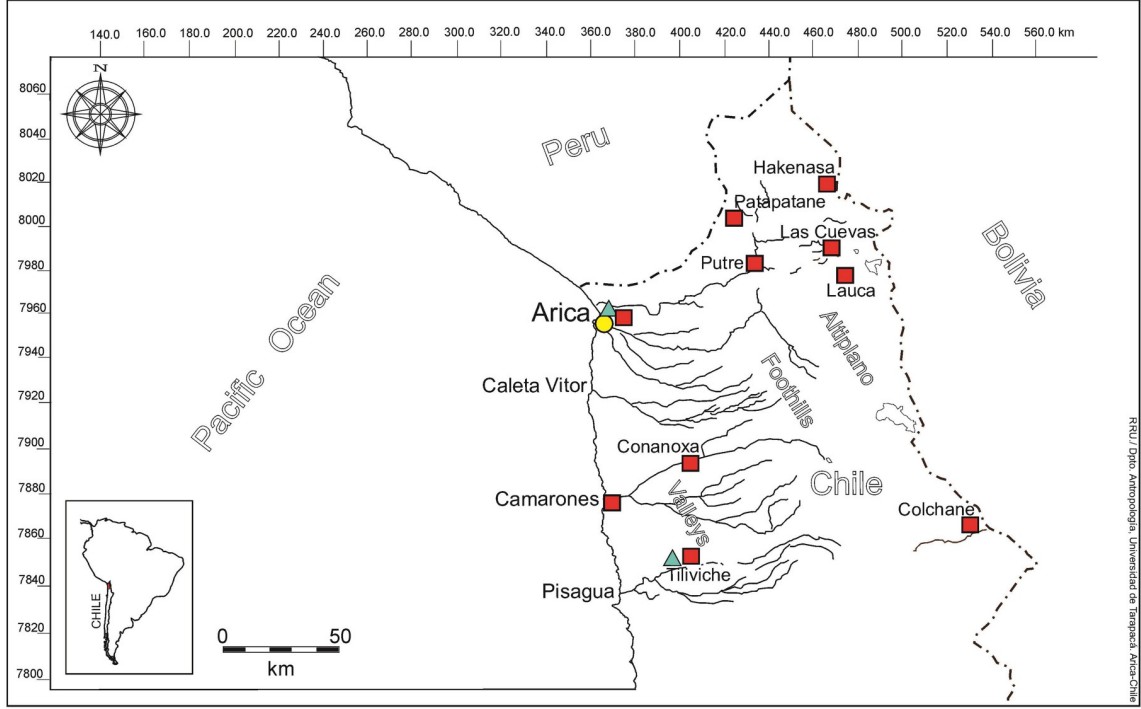

**Fig 1. Map of northern Chile.** Coastal territory inhabited by maritime populations of fishers, hunters and gatherers and sampling localities of fauna for strontium line base: wild rodents captured in the coast, valleys, foothills and altiplano (red squares); samples marine mammals (blue triangle); human samples from Arica coast (yellow circle, including individuals with and without marks of violence).

one estimates that violence and war were not constitutive of the social structure among hunters and gatherers, due to the lack of social and environmental condition to practice it, or to maintain antagonist groups in recurrent confrontation [3–7], add that lethal violence between individuals of small populations with high mobile residence, could have jeopardized their survival. Conversely, collective violence and war would have been a socio-cultural feature of farming societies, concerned with the defense of capital investment (irrigation canal, dams, different form of land preparation, silviculture). Besides defense and protection, in some cases, there was the need for territorial expansion, which was eventually solved by conflicting and competing with foreign social groups [4, 8]. In sum, violence would have increase through time linked to social complexity and population increase. Violence in the ecological perspective stems from the availability of subsistence resources, becoming an adaptive response to deal with their environmental fluctuation [9–13]. Environmental stress coupled with scarcity of subsistence resources should trigger competition and hostile behavior among human groups [12]. In contrast, buoyance periods, with greater availability of resources, conflicts and violence would diminish, and banquet and other social expression (funeral and festive ceremonies) could have taken place [12].

For the evolutionary perspective, violence and war would be anchored in the biology of our species and shaped through its evolution, which would mean that this behavior would have accompanied *Homo sapiens* in its long and complex evolutionary trajectory [7, 10–12, 14–19]. Thus, the capacity for aggressive behavior would be inherent to the human condition, and its evolution would be shaped by natural selection [e.g., 15, 20, 21]. In this theoretical view, specific patterns of violence would be generated and shaped by environmental and sociocultural factors. These factors would have a greater or lesser influence on the existence, intensity and effects of violence in people's lives.

For the bioarchaeological perspective and social theory, violence and war would be codified by intricate cultural meanings [20–22], which implies that social and symbolic factors would channel different forms of violence, from their own cultural logics, which can be identified and interpreted from the skeletal remains. For prehistoric populations that governed their lives without written records, certain behaviors that from a Western perspective can be seen as violent, irrational, aberrant or deviant, could have had totally different cultural meanings. In other words, violence should not be considered a simple reaction or response to external factors (e.g., environmental stress, competition for resources, population growth), nor would it derive from an innate human biological condition.

Social and cultural contexts give violence its power and meaning, which means that different forms of violence would be tolerated, and its use would be culturally legitimized for social control and stability. These approaches have argued that functionalist models of violence, such as those discussed in the preceding paragraphs, limit interpretations of violence in its social and symbolic dimensions [see 14, 22–25].

From the perspective of historical anthropology, Ferguson and Whitehead [3], proposed the thesis of "tribal zone warfare" for territories or zones of conquest and subjugation by state or imperial societies over politically decentralized native societies. Contact between dominators and subjugated in the tribal zone would have caused a substantial intensification of warfare and violence, due to the introduction of new steel weapons of greater lethality, altering the cultural patterns of native societies. Ferguson and Whitehead [3] applied this model to understand and interpret the historically and ethnographically documented warfare in the Amazonian lowlands of South America. Our study is also situated in the South American lowlands, specifically in the narrow coastal strip along the Atacama Desert, one of the driest on the planet. The populations studied correspond small-scale and politically decentralized societies, with a scale of 10,000 years of trajectory and prior to contact with the West.

In a previous study, and using a bioarchaeological approach [26] studied violence in populations of hunters, fishers, and gatherers from the Archaic Period of the Chinchorro culture that lived in the Atacama Desert between 10,000 and 4000 cal BP [27]. The results revealed that about 25% of the individuals analyzed showed signs or traces of trauma compatible with interpersonal violence. One unexpected aspect of this study is that it shows us that violence remained constant through time, from periods ranging from the Early Archaic (10,000 cal BP) to the Late Archaic (4000 cal BP). Thus, for 6000 years, violence among these coastal populations was invariant and, therefore, independent of demographic and ecological factors. With this background, it will be relevant to know the trajectory of violence among coastal populations, which maintained the same subsistence mode of fishing, hunting, and gathering, during the periods after the Archaic, i.e., the Formative (3000–1500 BP) and Late Intermediate (AD 1000–1500) Periods. This will allow us to verify whether these patterns of violence were constant, or conversely, varied over a timescale of 10,000 years.

During the Formative Period, despite the introduction of horticulture in the Atacama Desert, not all groups left the coast. Some remained living along the shoreline, subsisting on coastal resources, maintaining and perfecting the millennia-old technologies of marine fishing and hunting. Others began experimenting with imported plants, developing horticulture on the river terraces of the narrow valleys and quebradas that drain down from the Andean Mountains towards the Pacific Ocean [28]. Thus, during this Period, two relevant events occurred that could have influenced or triggered violence among the formative populations. On the one hand, neighborhoods emerged as permanent populations settled in the valleys adjacent to the coast to cultivate the scarce productive spaces there, i.e., the coastal populations were no longer alone, but now had close neighbors. On the other hand, the exchange networks intensified via caravans of camelids (llamas), which had already become fully domesticated in the Andean highlands [29]. These pack animals allowed for the circulation of prestige goods (e.g., camelid wool and other sumptuary goods) throughout the south central Andes region, including the Atacama Desert and the Pacific coast, which may have generated disputes and quarrels over access to these goods by fishing and horticultural groups.

During the Late Intermediate Period (LIP), a change occurred in the social organization of these fishing populations: they became politically and socially organized into small independent and autonomous chiefdoms of fishers, who controlled the coastal edge, and farmers, who controlled the adjacent valleys, both groups being independent of one another [30–32]. One factor that would have favored the increase in violence over time would have been the multiethnic occupation that has been proposed for the coast of the Atacama Desert, where several ethnic groups would have shared and exploited the same coastal spaces. This multiethnicity could have generated friction and competition between groups for control and access to safer and more easily accessible coastal sites to exploit fishery resources. Alternatively, these later populations with greater social complexity could have implemented mechanisms of social order that would have exercised a certain hierarchical control, inhibiting or minimizing violent behavior.

Against this backdrop, the long record of human social history in northern Chile offers an unparalleled opportunity to test alternative hypotheses regarding violence. In particular, we ask whether violence among these fishing, hunting, and gathering populations would have decreased over time from the Archaic to the Late Intermediate Periods, given the greater social complexity of the later populations; or conversely, did violence increased over time, given the multiethnicity and competition for the same resources? Another possible scenario is that violence had been fluctuating over time and have been related to periods of environmental stress (e.g., ENSO linked with dry periods, or summer torrential downpours in the altiplano link to La Niña period) [33, 34]. In addition, we assessed whether interpersonal violence affected

males and females equally over time or, conversely, whether violence affected one sex more than the other. A last scenario could show that violence have been invariant over time, as it was for the Archaic Period [26]. These possible trajectories were contrasted with other socio-cultural manifestations, such as rock art, weapons in mortuary contexts and settlement patterns, as expressions of violent behavior. In other words, violence as a social phenomenon can be observed not only through the traces left by confrontations on the human body, but also through other expressions of material culture such as those mentioned above.

## Material and methods

### The bioarchaeological sample

A total of 288 adult individuals were examined for evidence of trauma caused by interpersonal violence (S1 Table). These collections were recovered from 20 funerary sites excavated during the last 50 years along the coast of Arica in the extreme north of Chile (Fig 2). All of individuals (except collection PlM-9) are stored at the Museo de Arqueología de la Universidad de Tarapacá in San Miguel de Azapa (MASMA), Arica, Chile. This coastal human sample constitutes a first solid corpus of data to assess long-term violence in fishing, hunting, and gathering societies of the Atacama Desert. Some sites are represented by more than fifty individuals, while

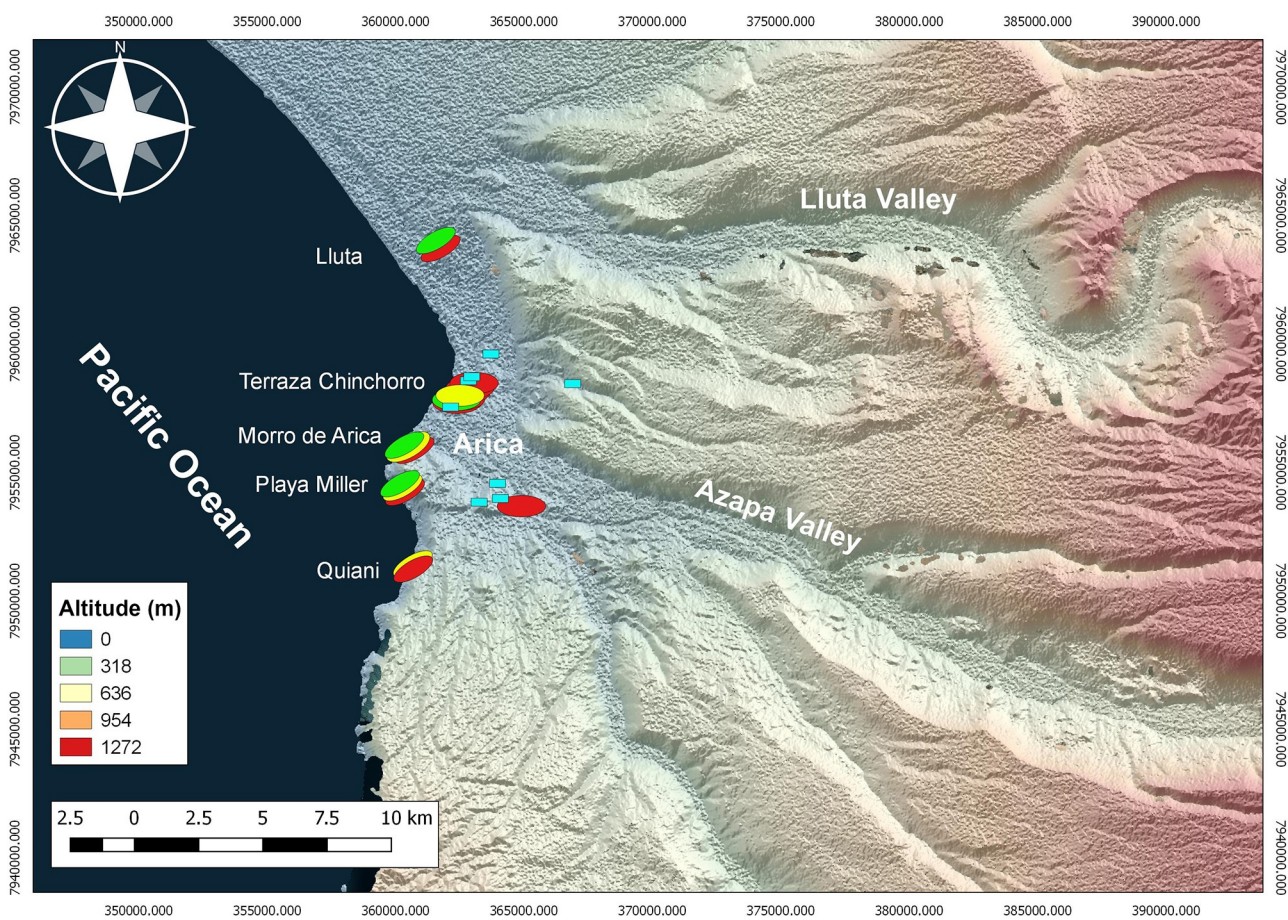

**Fig 2. Location of maritime populations on the Arica coast.** (shell midden and burial sites): Red dots, Archaic sites (Acha, Quiani, Playa Miller, Morro de Arica, Terraza Chinchorro, Lluta). Yellow dots, Formative sites (Quiani, Playa Miller, Morro de Arica). Green dots, Late Intermediate Period sites (Playa Miller, Morro de Arica). Turquoise dots, isolated Formative Period inhumations.

**Table 1. Sites and number of individuals from the Arica coast included in this study.** Archaic Period (10,000–4000 BP); Formative Period (3000–1500 BP); Late Intermediate Period (1000–550 BP).

| Site | N | M | F | Period | Chronology cal BP | Coast | Reference[*] |
|---|---|---|---|---|---|---|---|
| Acha | 4 | 3 | 1 | Early Archaic | 10,000–8000 | Pampa | [39] |
| Playa Miller-8 | 13 | 8 | 5 | Middle and Late Archaic | 6500–4000 | South | [26] |
| Morro-1 | 68 | 34 | 34 | Middle and Late Archaic | 6000–4000 | South | [26] |
| Morro-1/6 | 35 | 20 | 15 | Late Archaic | 4500–4000 | South | [40] |
| Morro-1/5 | 3 | 1 | 2 | Late Archaic | 4500–4000 | South | [26] |
| Morro Cárcel | 1 | 1 | 0 | Late Archaic | 4000 | South | [26] |
| Bolognesi | 3 | 1 | 2 | Late Archaic | 4000 | South | [26] |
| Colón 10 | 1 | 1 | 0 | Late Archaic | 4000 | South | [26] |
| Maestranza CH | 5 | 2 | 3 | Middle Archaic | 6000 | North | [39] |
| Maderas Enco | 2 | 1 | 1 | Middle Archaic | 5000 | North | [35] |
| Mina Macarena | 1 | 1 | 0 | Middle Archaic | 5000 | North | [26] |
| Quiani-7 | 9 | 4 | 5 | Early Formative | 3000–2800 | South | [**] |
| Morro-1/6D | 9 | 3 | 6 | Early Formative | 2800–2500 | South | [40] |
| Ejército 163 | 1 | 1 | 0 | Early Formative | 2800 | South | [**] |
| Morro-2/2 | 1 | 1 | 0 | Late Formative | 2800 | South | [41] |
| Playa Miller-7 | 61 | 30 | 31 | Late Formative | 2500–1500 | South | [37] |
| Garibaldi | 1 | 0 | 1 | Late Formative | 2000 | Pampa | [**] |
| Playa Miller-3 | 2 | 2 | 0 | Late Intermediate Period | 1000–450 | South | [37] |
| Playa Miller-4 | 45 | 27 | 18 | Late Intermediate Period | 1000–450 | South | [42] |
| Playa Miller-9 [***] | 23 | 14 | 9 | Late Intermediate Period | 1000–450 | South | [37] |
| **Total** | **288** | **155** | **133** | | | | |

N = number of infividual; M = male; F = female

[*] It corresponds to the archaeological reports of the funerary sites.

[**] Unpusblished reports on files stored at Laboratorio de Antropología Física, Museo de Arqueología de la Universidad de Tarapacá, San Miguel de Azapa (MASMA), Arica, Chile.

[***] This collection was studied by one of the co-authors (B. Arriaza 1997), but it is currently unavailable since the bodies were reburied in the 1990s.

others by only a couple of individuals (Table 1). The bodies show different degrees of preservation, mostly skeletons, but some of them preserved soft tissues (approximately 20%). As regards the degree of completeness, some bodies are complete, and others represented only by skulls or postcranial bones. The sites cover a chronology ranging from the Early Archaic (10,000 cal BP) [26, 35, 36]) to the Late Intermediate Periods (AD 1500) [37, 38]). The total sample of individuals was separated in two subcategories according to their degree of completeness: (a) individuals with only the skull preserved, and (b) individuals with both skull and postcranial preserved. The first subcategory corresponds only to skulls with trauma; the second category, counted independently of the first, corresponds to postcranial trauma, regardless of its completeness. Additionally, individuals with soft tissue remains (ca. 20% of the total sample studied), allowed us to visualize traumas that only affected these tissues and were added to the total number of individuals with trauma analyzed. Some of the cases with more complete soft tissue were X-rayed, which did not show bone trauma, since cutaneous and subcutaneous wounds in areas such as the abdomen do not leave imprints on the bones.

**Age and sex.** The methods and techniques to estimate age and sex of the individuals were the usually empleaded in the bioarchaeological analysis [43]. For determinate the sex, this facilitated by some individuals with their external genitalia still preserved. In this work, individuals >17 years old were recorded as adults, of which 154 were male and 134 were female.

## Bioarchaeology and interpersonal violence traumas

Traumas embodied in corpses are the most direct evidence of human experiences of conflict and social tension in the past. These violent behaviors that caused physical harm to another individual may be expressed in trauma at the level of hard tissues such as the skeleton and/or soft tissues such as the skin, organs, and scalp. However, given the difference in the preservation of both types of tissues, bioarchaeological studies on interpersonal violence have focused almost exclusively on the study of skeletal trauma [11, 24, 25, 44–51]. This forced choice–of bone tissue—introduces an inherent bias into the study of violence in the past. This has culminated in an important methodological limitation, because a considerable number of traumas will affect only soft tissues, which for conservation reasons remain invisible and have therefore been scarcely reported in studies on violence in the past [52]. On the other hand, it is through the study of bodies showing signs of trauma that we can learn about the type of violence exerted between individuals or groups of individuals, the effects of violent behaviors, the forms of fighting and brawling, as well as the weapons that may have been used to cause such injuries [11, 25]. By contrast, the motivations or causes as to why human societies choose to perpetrate physically violent behaviors, causing certain individuals harm and suffering, are too complex to be able to infer from the bioarchaeological record, even more so for societies that left no written record, such as the millennia-old Andean populations.

In general, traumas interpreted as expression of interpersonal violence correspond to those affecting the axial skeleton, such as the skull and thorax, and those affecting the bones of the upper appendicular skeleton, such as the classic parry fracture in the diaphysis of the ulna, interpreted as a fending fracture or blockage of a blow to the head or face [53]. The projectile points embedded in the bones is another indicator of violence, mainly interpreted as inter-group violence [25, 45, 54]. When the projectile point is embedded in the bone and there are no signs of bone regeneration, it is inferred that the trauma was perimortem, with lethal consequences. By contrast, the presence of a projectile not embedded in some bone of the skeletal, could be interpreted as an offering, masking its primary position where it may have impacted some organ of the trunk cavities, leaving no trace in the skeleton of the cause of death.

The good preservation of the bioanthropological remains analyzed facilitated the identification of antemortem and perimortem traumas, especially bone fractures. Antemortem fractures are the most studied in the bioarchaeological record, because most of them leave clear evidence of the bone's healing process. Depressed fractures of the cranial vault are a good indicator of interpersonal violence and are expressed according to their severity as a sinking or "dent" in the outer table of the bones [11]. Facial bones are also highly sensitive to interpersonal violence trauma, affecting mostly nasal, zygomatic, and maxillary bones. In the long bones of the upper extremity, when the diaphysis is affected, it shows traces of bone callus formation due to a healed fracture. Rib fractures have also been identified as indicative of direct blows to the trunk [26], although they can also result from falls in the context of brawls and fights, or simply an accidental fall.

In the case of perimortem trauma, given that the bone is still fresh and retains its moisture, the pattern of the fracture can be differentiated from postmortem taphonomic fractures, i.e., when the bone is already dry [55]. In the analyzed collections, the characteristics or indicators that were considered as diagnostic elements for the classification of perimortem trauma were: beveled edges of the fractured bones, homogeneity of color of the fractured surface with that of the rest of the bone, hinged bone fragments and radial fractures in the skull [55]. A relevant factor in the bioanthropological collections studied is that about 20% of the individuals preserved soft tissue remains, facilitating more accurate diagnoses as the bone fragments of the

fractures are maintained in situ. Thus, it was possible to observe the resulting deformation in the area where the high impact blow was received that caused the damage.

This range of traumas provided the key indicators for determining the type of violence exerted on these fishing, hunting, and gathering populations adapted to the Atacama Desert's maritime ecosystem.

## Geoarchaeology: Strontium (Sr) isotopes

Geoarchaeology, in this case using strontium isotopes $^{87}$Sr/$^{86}$Sr as a proxy, has been incorporated into the studies of violence to measure the chemical signal of an individual and to infer if they were local or foreign in origin. This may provide an indication as to whether the violence had been exercised at the intra-group level or between factions or local groups living in and sharing the same territory; or conversely, if they had come from other geological localities. The latter scenario could be interpreted as indicative of inter-group violence.

**Sampling localities.** The Sr isotopic baseline was elaborated from the animal model for the extreme north of Chile (~18˚30'-19˚40'S). Bone samples of modern and archaeological fauna, both marine and terrestrial, were taken from 10 localities located from sea level to 4,200 masl (Fig 1). Modern wild rodent fauna (n = 12) were captured with Sherman traps that were placed in uninhabited areas to ensure that rodents had exclusively consumed local grasses. They were captured in: (a) the mouths of the Arica and Camarones rivers, 2 km from the current beach line and at 0 masl; (b) the desert oasis (Tiliviche and Conanoxa) 30–40 km from the coast respectively and at 900 masl; foothills (Patapatane and Putre) 70 km from the coast and at 3,700 masl; and the altiplano (Hakenasa, Las Cuevas, Lauca, Colchane) between 4,000–4,200 masl and 130–150 km from the coast (Fig 1). In addition, camelid bone samples (n = 6) were collected at six Archaic archaeological sites, distributed in the same area where the modern fauna samples were taken (Fig 1; Table 2). The marine mammal bone samples (n = 3), correspond to a sea lion found dead on the shore of the beach (Arica coast) and two archaeological samples (whale rib from the Arica coast; and sea lion from Tiliviche (which must have been moved from the coast to the interior oasis by Archaic populations).

We then proceeded to sample 108 individuals, with and without traces of trauma from interpersonal violence, from which we obtained dental enamel (3 mg) or bone (3 g) samples. To the 56 samples already known for the Archaic Period [26], 52 new samples were added, of which 39 correspond to the Formative Period and 13 to the LIP. With this proxy, we evaluated whether the individuals buried on the coast of Arica that showed traces of violence presented a local chemical signal, i.e., marine, which would suggest tensions between local groups. By contrast, if the individuals with traumas presented a non-marine signature, this would indicate violent actions with actors from other localities or regions bordering the desert, such as the valleys of Arica or southern Peru, the altiplano, and even the eastern lowlands of tropical forest.

**Processing the samples.** All samples for Sr isotope geochemistry were processed in the radiogenic isotope geochemistry laboratory at the University of North Carolina—Chapel Hill. Superficial contamination on samples was removed by ultrasound cleaning and light polishing with a motorized polishing bit. The same tool with a diamond bit was used to powder 2–3 mg of tooth enamel (human samples) or bone (animals). All samples were dissolved in 3.5 M $HNO_3$ and Sr was isolated from the samples using standard column chromatographic techniques. One microliter of concentrated $H_3PO_4$ was added to the Sr solution and the samples were evaporated to a small drop for analysis. All samples were loaded onto single Re filaments with $TaCl_5$ and loaded into the VG-Sector-54 thermal ionization mass spectrometer at Chapel Hill. Analyses were done using triple-dynamic multicollector mode with $^{88}$Sr—3 V (10–11 Ω resistor). Data data were normalized to $^{86}$Sr/88Sr = 0.1194 assuming exponential mass

**Table 2. Strontium isotope ratios in human teeth from the coastal populations of Arica and animal bones from northern Chile (coast, valleys, altiplano).**

| Site | ID Body | $^{87}Sr/^{86}Sr^1$ | Trauma | Sample type | Age | Sex | Region | Period | Culture | Human/faunal |
|---|---|---|---|---|---|---|---|---|---|---|
| Acha 3 | Acha3C1 | 0.708703 | Trauma | $M_3$ left | Adult | M | Arica Coast | Early Archaic | Chinchorro | Human |
| Acha 3 | Acha3C4 | 0.708607 | Trauma | $M^3$ left | Adult | M | Arica Coast | Early Archaic | Chinchorro | Human |
| Morro 1 | M1T1C4 | 0.708876 | Trauma | $M_3$ right | Adult | M | Arica Coast | Middle Archaic | Chinchorro | Human |
| Morro 1 | M1T28C22 | 0.708922 | Trauma | $M^3$ right | Adult | M | Arica Coast | Late Archaic | Chinchorro | Human |
| Morro 1 | M1T28C3 | 0.708497 | Trauma | $M^3$ left | Adult | M | Arica Coast | Late Archaic | Chinchorro | Human |
| Morro 1 | M1R1 | 0.708866 | Trauma | $M_2$ right | Adult | M | Arica Coast | Late Archaic | Chinchorro | Human |
| Morro 1 | M1T031B | 0.708952 | Trauma | $M^1$ right | Adult | F | Arica Coast | Late Archaic | Chinchorro | Human |
| Morro 1 | M1T16B | 0.708558 | Trauma | $M^3$ right | Adult | M | Arica Coast | Late Archaic | Chinchorro | Human |
| Morro 1 | M1T27C18 | 0.706385 | Trauma | $M_2$ right | Adult | M | Arica Coast | Late Archaic | Chinchorro | Human |
| Morro 1 | M1T19C1 | 0.708848 | Trauma | $M_2$ right | Adult | M | Arica Coast | Late Archaic | Chinchorro | Human |
| Morro 1 | M1T22C5a | 0.708918 | Trauma | $M_3$ left | Adult | M | Arica Coast | Late Archaic | Chinchorro | Human |
| Morro 1 | M1T23C4 | 0.708719 | Trauma | $M^3$ right | Adult | F | Arica Coast | Late Archaic | Chinchorro | Human |
| Morro 1 | M1T27C13 | 0.708939 | Trauma | Rib | Adult | F | Arica Coast | Late Archaic | Chinchorro | Human |
| Morro 1 | M1T27C8 | 0.708903 | Trauma | $M^3$ left | Adult | F | Arica Coast | Late Archaic | Chinchorro | Human |
| Morro 1 | M1T28C13 | 0.709034 | Trauma | $M_3$ left | Adult | M | Arica Coast | Late Archaic | Chinchorro | Human |
| Morro 1 | M1T28C2 | 0.708913 | Trauma | Enamel | Adult | F | Arica Coast | Late Archaic | Chinchorro | Human |
| Morro 1 | M1T28C9 | 0.708980 | Trauma | $M^2$ left | Adult | F | Arica Coast | Late Archaic | Chinchorro | Human |
| Morro 1 | M1T8 | 0.708731 | Trauma | Rib | Adult | F | Arica Coast | Late Archaic | Chinchorro | Human |
| Morro 1/6 | M1/6T10A | 0.708872 | Trauma | $PM^2$ right | Adult | F | Arica Coast | Late Archaic | Chinchorro | Human |
| Morro 1/6 | M1/6T5 | 0.708895 | Trauma | Rib | Adult | F | Arica Coast | Late Archaic | Chinchorro | Human |
| Morro 1/6 | M1/6T44 | 0.708604 | Trauma | $M_3$ left | Adult | M | Arica Coast | Late Archaic | Chinchorro | Human |
| Morro 1/6 | M1/6T4N | 0.708816 | Trauma | $M_3$ left | Adult | M | Arica Coast | Late Archaic | Chinchorro | Human |
| Morro 1/6 | M1/6T63 | 0.708869 | Trauma | $M_3$ right | Adult | M | Arica Coast | Late Archaic | Chinchorro | Human |
| Morro 1/6 | M1/6T10 | 0.708885 | Trauma | $PM^2$ right | Adult | F | Arica Coast | Late Archaic | Chinchorro | Human |
| Morro 1/6 | M1/6T32 | 0.708850 | Trauma | $M_3$ right | Adult | F | Arica Coast | Late Archaic | Chinchorro | Human |
| Morro 1/6 | M1/6TUI | 0.708792 | Trauma | Rib | Adult | F | Arica Coast | Late Archaic | Chinchorro | Human |
| Morro 1/6 | M1/6T18 | 0.708790 | Trauma | $M_3$ left | Adult | M | Arica Coast | Late Archaic | Chinchorro | Human |
| Morro 1/6 | M1/6T22 | 0.708826 | Trauma | $PM_2$ right | Adult | M | Arica Coast | Late Archaic | Chinchorro | Human |
| Morro C/T0 | M/CT0 | 0.708763 | Trauma | $I_2$ right | Adult | M | Arica Coast | Late Archaic | Chinchorro | Human |
| Bolognesi | Bolognesi_P1C1 | 0.709007 | Trauma | Rib | Adult | M | Arica Coast | Late Archaic | Chinchorro | Human |
| Maderas Enco | MEC1 | 0.708743 | Trauma | Vertebrae | Juvenile | M | Arica Coast | Late Archaic | Chinchorro | Human |
| Acha 3 | Acha3C2 | 0.708632 | Negative | $M_1$ left | Infant | U | Arica Coast | Early Archaic | Chinchorro | Human |
| Acha 4 | Acha4C1 | 0.708807 | Negative | $M^3$ right | Adult | F | Arica Coast | Early Archaic | Chinchorro | Human |
| Morro 1 | M1T23C12 | 0.708752 | Negative | $M_3$ left | Adult | F | Arica Coast | Late Archaic | Chinchorro | Human |
| Morro 1 | M1T25C6 | 0.708712 | Negative | $M^1$ left | Infant | U | Arica Coast | Late Archaic | Chinchorro | Human |
| Morro 1 | M1T22C5 | 0.708640 | Negative | $M_3$ right | Adult | F | Arica Coast | Late Archaic | Chinchorro | Human |
| Morro 1 | M1T28C24 | 0.708876 | Negative | $PM^2$ left | Adult | M | Arica Coast | Late Archaic | Chinchorro | Human |
| Morro 1/5 | M1/5TIII | 0.708866 | Negative | $M_1$ right | Infant | M | Arica Coast | Late Archaic | Chinchorro | Human |
| Morro 1/5 | M1/5TXII | 0.708662 | Negative | $M_1$ right | Infant | U | Arica Coast | Late Archaic | Chinchorro | Human |
| Morro 1/6 | M1/6T28 | 0.708967 | Negative | $M_2$ right | Infant | M | Arica Coast | Late Archaic | Chinchorro | Human |
| Colon 10 | Col-10Cal-1 | 0.708745 | Negative | $M^2$ right | Adult | M | Arica Coast | Late Archaic | Chinchorro | Human |
| Playa Miller 8 | PLM8TC | 0.708955 | Negative | $M_2$ right | Adult | F | Arica Coast | Late Archaic | Chinchorro | Human |
| Playa Miller 8 | PLM8CR2 | 0.709059 | Negative | $M^2$ left | Adult | M | Arica Coast | Late Archaic | Chinchorro | Human |
| Playa Miller 8 | PLM8CR01 | 0.709024 | Negative | $PM_2$ right | Adult | M | Arica Coast | Middle Archaic | Chinchorro | Human |

(*Continued*)

**Table 2.** (Continued)

| Site | ID Body | $^{87}Sr/^{86}Sr^1$ | Trauma | Sample type | Age | Sex | Region | Period | Culture | Human/faunal |
|---|---|---|---|---|---|---|---|---|---|---|
| Playa Miller 8 | PLM8B1 | 0.709061 | Negative | $PM^2$ right | Adult | F | Arica Coast | Middle Archaic | Chinchorro | Human |
| Playa Miller 8 | PLM8T4 | 0.708800 | Negative | $M^1$ left | Infant | U | Arica Coast | Late Archaic | Chinchorro | Human |
| Playa Miller 8 | PLM8Cr8 | 0.708922 | Negative | $M^1$ right | Adult | F | Arica Coast | Late Archaic | Chinchorro | Human |
| Playa Miller 8 | PLM8Cr14 | 0.708855 | Negative | $M_3$ right | Adult | M | Arica Coast | Late Archaic | Chinchorro | Human |
| Playa Miller 8 | PLM8Cr30 | 0.708934 | Negative | $M_2$ right | Adult | F | Arica Coast | Late Archaic | Chinchorro | Human |
| Playa Miller 8 | PLM8Cr6 | 0.708789 | Negative | $M^2$ right | Adult | F | Arica Coast | Late Archaic | Chinchorro | Human |
| Playa Miller 8 | PLM8T4Cr2e | 0.708893 | Negative | $M^2$ left | Adult | U | Arica Coast | Late Archaic | Chinchorro | Human |
| Playa Miller 8 | PLM8T4Cr2 | 0.709017 | Negative | $M^2$ right | Adult | M | Arica Coast | Late Archaic | Chinchorro | Human |
| Chinchorro 1 | CH1C1 | 0.708868 | Negative | $M_2$ right | Infant | U | Arica Coast | Middle Archaic | Chinchorro | Human |
| Maestranza CH | MtzaCH-C1 | 0.708887 | Negative | $M_2$ right | Adult | F | Arica Coast | Middle Archaic | Chinchorro | Human |
| Maestranza CH | MtzaCH-C9 | 0.708898 | Negative | $M^2$ right | Adult | F | Arica Coast | Middle Archaic | Chinchorro | Human |
| Maestranza CH | MtzaCH-C10 | 0.708926 | Negative | $M^2$ left | Adult | M | Arica Coast | Middle Archaic | Chinchorro | Human |
| Quiani 7 | QITA | 0.708863 | Trauma | $M^3$ right | Adult | M | Arica Coast | Early Formative | Quiani | Human |
| Quiani 7 | QIT13 | 0.708792 | Trauma | $M^3$ left | Adult | F | Arica Coast | Early Formative | Quiani | Human |
| Morro 1-6D | M1/6DT10 | 0.708746 | Trauma | $M_2$ left | Adult | M | Arica Coast | Early Formative | Faldas del Morro | Human |
| Ejército 163 | EJ163C1 | 0.708636 | Trauma | $M_2$ right | Adult | M | Arica Coast | Early Formative | Faldas del Morro | Human |
| Garibaldi | Garibaldi C1 | 0.708824 | Trauma | Bone | Adult | F | Arica Coast | Late Formative | Formative | Human |
| Playa Miller 7 | PLM7CR5A | 0.708710 | Trauma | $M^2$ right | Adult | F | Arica Coast | Late Formative | El Laucho | Human |
| Playa Miller 7 | PLM7CR8A | 0.708693 | Trauma | $M^3$ left | Adult | M | Arica Coast | Late Formative | El Laucho | Human |
| Playa Miller 7 | PLM7CR9 | 0.708383 | Trauma | $M^1$ left | Adult | M | Arica Coast | Late Formative | El Laucho | Human |
| Playa Miller 7 | PLM7CR10 | 0.708613 | Trauma | Cranium | Adult | M | Arica Coast | Late Formative | El Laucho | Human |
| Playa Miller 7 | PLM7CR12 | 0.708819 | Trauma | $M^3$ right | Adult | M | Arica Coast | Late Formative | El Laucho | Human |
| Playa Miller 7 | PLM7CR22 | 0.708884 | Trauma | Cranium | Adult | F | Arica Coast | Late Formative | El Laucho | Human |
| Playa Miller 7 | PLM7T24 | 0.707281 | Trauma | $M_3$ left | Adult | M | Arica Coast | Late Formative | El Laucho | Human |
| Playa Miller 7 | PLM7CR100 | 0.708688 | Trauma | $M^3$ left | Adult | M | Arica Coast | Late Formative | El Laucho | Human |
| Playa Miller 7 | PLM7T139 | 0.708853 | Trauma | $I_2$ left | Adult | F | Arica Coast | Late Formative | El Laucho | Human |
| Playa Miller 7 | PLM7CR144 | 0.708757 | Trauma | $M^2$ right | Adult | F | Arica Coast | Late Formative | El Laucho | Human |
| Playa Miller 7 | PLM7T328 | 0.708765 | Trauma | $M^2$ left | Adult | F | Arica Coast | Late Formative | El Laucho | Human |
| Playa Miller 7 | PLM7/Ex4/Sn/1 | 0.707420 | Trauma | $M_3$ right | Adult | M | Arica Coast | Late Formative | El Laucho | Human |
| Quiani 7 | QIT12 | 0.708814 | Negative | Rib | Adult | M | Arica Coast | Early Formative | Quiani | Human |
| Quiani 7 | QIT15 | 0.709176 | Negative | Vertebrae | Infant | U | Arica Coast | Early Formative | Quiani | Human |
| Quiani 7 | QIT16 | 0.708795 | Negative | $M_2$ right | Adult | F | Arica Coast | Early Formative | Quiani | Human |
| Quiani 7 | QIT16A | 0.708735 | Negative | Rib | Adult | F | Arica Coast | Early Formative | Quiani | Human |
| Quiani 7 | QIT17 | 0.708686 | Negative | $M^3$ left | Adult | F | Arica Coast | Early Formative | Quiani | Human |
| Quiani 7 | QIT17A | 0.708874 | Negative | Vertebrae | Infant | U | Arica Coast | Early Formative | Quiani | Human |

*(Continued)*

**Table 2.** (*Continued*)

| Site | ID Body | $^{87}Sr/^{86}Sr^1$ | Trauma | Sample type | Age | Sex | Region | Period | Culture | Human/faunal |
|---|---|---|---|---|---|---|---|---|---|---|
| Quiani 7 | QIT18 | 0.708913 | Negative | Vertebrae | Infant | U | Arica Coast | Early Formative | Quiani | Human |
| Quiani 7 | QIT19 | 0.708767 | Negative | Phalanx | Infant | M | Arica Coast | Early Formative | Quiani | Human |
| Quiani 7 | QIT21 | 0.708842 | Negative | Rib | Adult | M | Arica Coast | Early Formative | Quiani | Human |
| Quiani 7 | QI25/07/05 | 0.708883 | Negative | $M^3$ left | Adult | U | Arica Coast | Early Formative | Quiani | Human |
| Quiani 7 | QI30/12/97 | 0.708467 | Negative | $M_1$ left | Adult | U | Arica Coast | Early Formative | Quiani | Human |
| Morro 1-6D | M1/6DT1 | 0.708542 | Negative | Rib | Adult | F | Arica Coast | Early Formative | Faldas del Morro | Human |
| Morro 1-6D | M1/6DT2 | 0.708511 | Negative | $M_3$ left | Adult | F | Arica Coast | Early Formative | Faldas del Morro | Human |
| Morro 1-6D | M1/6DT2A | 0.708677 | Negative | Bone | Infant | U | Arica Coast | Early Formative | Faldas del Morro | Human |
| Morro 1-6D | M1/6DT3 | 0.708973 | Negative | Rib | Juvenile | M | Arica Coast | Early Formative | Faldas del Morro | Human |
| Morro 1-6D | M1/6DT4 | 0.708759 | Negative | Rib | Adult | F | Arica Coast | Early Formative | Faldas del Morro | Human |
| Morro 1-6D | M1/6DT5 | 0.708663 | Negative | Rib | Adult | F | Arica Coast | Early Formative | Faldas del Morro | Human |
| Morro 1-6D | M1/6DT6 | 0.708919 | Negative | Rib | Adult | F | Arica Coast | Early Formative | Faldas del Morro | Human |
| Morro 1-6D | M1/6DT7 | 0.708789 | Negative | Rib | Adult | M | Arica Coast | Early Formative | Faldas del Morro | Human |
| Morro 1-6D | M1/6DT8 | 0.708661 | Negative | $M_2$ left | Adult | F | Arica Coast | Early Formative | Faldas del Morro | Human |
| Morro 1-6D | M1/6DT9 | 0.708788 | Negative | $M_3$ left | Adult | F | Arica Coast | Early Formative | Faldas del Morro | Human |
| Playa Miller 7 | PLM7CR15 | 0.708863 | Negative | $M^1$ right | Adult | M | Arica Coast | Late Formative | El Laucho | Human |
| Playa Miller 4 | PLM4T159 | 0.708665 | Trauma | $M_3$ left | Adult | M | Arica Coast | LIP | LIP | Human |
| Playa Miller 4 | PLM4SN3 | 0.708463 | Trauma | $M^1$ right | Adult | F | Arica Coast | LIP | LIP | Human |
| Playa Miller 4 | PLM4SN4 | 0.708315 | Trauma | $M_3$ right | Adult | M | Arica Coast | LIP | LIP | Human |
| Playa Miller 4 | PLM4/Ex7/CR01 | 0.707781 | Trauma | $M^3$ left | Adult | M | Arica Coast | LIP | LIP | Human |
| Playa Miller 4 | PLM4 Desc | 0.708603 | Negative | $M_1$ left | Juvenile | U | Arica Coast | LIP | LIP | Human |
| Playa Miller 4 | PLM4 T10 | 0.708172 | Negative | $M^2$ left | Adult | M | Arica Coast | LIP | LIP | Human |
| Playa Miller 4 | PLM4 T79A | 0.708318 | Negative | $M_3$ right | Adult | F | Arica Coast | LIP | LIP | Human |
| Playa Miller 4 | PLM4 T99 | 0.708194 | Negative | $M_2$ right | Adult | F | Arica Coast | LIP | LIP | Human |
| Playa Miller 4 | PLM4T79B | 0.708692 | Negative | $M_2$ right | Adult | M | Arica Coast | LIP | LIP | Human |
| Playa Miller 4 | PLM4 T153 | 0.708823 | Negative | $M^2$ left | Adult | M | Arica Coast | LIP | LIP | Human |
| Playa Miller 4 | PLM4 T180 | 0.708408 | Negative | $M^2$ right | Adult | F | Arica Coast | LIP | LIP | Human |
| Playa Miller 4 | PLM4 T202 | 0.708685 | Negative | $M_3$ left | Adult | F | Arica Coast | LIP | LIP | Human |
| Playa Miller 4 | PLM4 T205B | 0.707859 | Negative | $M^2$ left | Adult | F | Arica Coast | LIP | LIP | Human |
| Playa Chinchorro | PlayaCH_Act | 0.709136 | - | Bone | Sea lion | - | Arica Coast | Modern | - | marine fauna |
| Maestranza CH | MtzaCH_Arq | 0.709163 | - | Bone | Whale | - | Arica Coast | Middle Archaic | - | marine fauna |
| Tiliviche 1B | Til_Arq_a | 0.709150 | - | Bone | Sea lion | - | Interior Desert | Late Archaic | - | marine fauna |

(*Continued*)

**Table 2.** (Continued)

| Site | ID Body | $^{87}Sr/^{86}Sr^1$ | Trauma | Sample type | Age | Sex | Region | Period | Culture | Human/faunal |
|---|---|---|---|---|---|---|---|---|---|---|
| Lluta | Lluta_Act_a | 0.707395 | - | Bone | Rodent | - | Lluta Valley Coast | Modern | - | terrestrial fauna |
| Lluta | Lluta_Act_b | 0.707210 | - | Bone | Rodent | - | Lluta Valley Coast | Modern | - | terrestrial fauna |
| Camarones | CAM_Act | 0.707402 | - | Bone | Rodent | - | Camarones Valley Coast | Modern | - | terrestrial fauna |
| Conanoxa | Cxa_Act_a | 0.707027 | - | Bone | Rodent | - | Camarones Valley | Modern | - | terrestrial fauna |
| Conanoxa | Cxa_Act_b | 0.706582 | - | Bone | Rodent | - | Camarones Valley | Modern | - | terrestrial fauna |
| Tiliviche 1B | Til_Arq_b | 0.706536 | - | Bone | Rodent | - | Interior Desert | Late Archaic | Late Archaic | terrestrial fauna |
| Tiliviche | Til_Act_a | 0.706602 | - | Bone | Rodent | - | Interior Desert | Modern | - | terrestrial fauna |
| Tiliviche | Til_Act_b | 0.706577 | - | Bone | Rodent | - | Interior Desert | Modern | - | terrestrial fauna |
| Patapatane | Pata_Arq_a | 0.707104 | - | Bone | Camelid | - | Foothills | Late Archaic | Late Archaic | terrestrial fauna |
| Patapatane | Pata_Arq_b | 0.706798 | - | Bone | Camelid | - | Foothills | Late Archaic | Late Archaic | terrestrial fauna |
| Putre | Putre_Act_a | 0.706878 | - | Bone | Rodent | - | Foothills | Modern | - | terrestrial fauna |
| Putre | Putre_Act_b | 0.707006 | - | Bone | Rodent | - | Foothills | Modern | - | terrestrial fauna |
| Hakenasa | Hake_Arq_a | 0.706868 | - | Bone | Camelid | - | Altiplano | Late Archaic | Late Archaic | terrestrial fauna |
| Hakenasa | Hake_Arq_b | 0.706819 | - | Bone | Camelid | - | Altiplano | Late Archaic | Late Archaic | terrestrial fauna |
| Lauca | Lauca_Arq | 0.706920 | - | Bone | Camelid | - | Altiplano | Late Archaic | Late Archaic | terrestrial fauna |
| Las Cuevas | Cuevas_Arq | 0.706730 | - | Bone | Camelid | - | Altiplano | Late Archaic | Late Archaic | terrestrial fauna |
| Colchane | Colcha_Act_b | 0.706470 | - | Bone | Rodent | - | Altiplano | Modern | - | terrestrial fauna |
| Colchane | Colcha_Act_a | 0.706116 | - | Bone | Rodent | - | Altiplano | Modern | - | terrestrial fauna |

fractionation behavior. Data are reported relative to NBS-987 $^{87}Sr/^{86}Sr = 0.710250$. Replicate analyses of the standard over the period of this study yielded and uncertainty in the $^{87}Sr/^{86}Sr$ of ± 0.000010 (n = 12).

### Indirect evidence: Settlement patterns, weapons, rock art, and geoglyphs

To complement these lines of research—bioarchaeological and geochemical—we evaluated other types of evidence for which violence can be expressed indirectly, such as the weapons used for these purposes, settlement patterns, and scenes represented in rock art and geoglyphs [44, 56, 57].

The settlement pattern is key, together with defensive architectural elements (walls, ditches, palisades), or the location of dwelling sites situated in strategic places with natural defensive features, such as steep hillsides or the tops of hills. The occurrence of these features suggests

situations of tension and conflict among human groups [10, 44] and have been widely studied in the Andes [21, 44]. Likewise, our investigation also focused on the artifacts that are part of the funerary contexts, with the purpose of identifying and characterizing the potential objects that would have been designed based on a prototype of a weapon, both offensive and defensive. As Walker [11] points out, the type of trauma identified can shed light on the weapon used to cause such injury to the body. On the other hand, we explore the presence of possible symbolic representations of confrontations and conflicts between groups, as an expression of the cultural dimension of violence in the extreme north of Chile, and to evaluate their correlation with the bioarchaeological evidence, we analyzed the images displayed in rock art (engravings and paintings) and geoglyphs located along the coast and valleys (up to 2000 m above sea level) and ascribed to the Formative and LIP Periods. A total of 60 rock engraving and painting sites, and 30 geoglyphs, inventoried to date [58], were reviewed with the aim of identifying scenes that included anthropomorphic motifs in a dynamic attitude of confrontation, carrying weapons, or clad in protective clothes, such as helmets or breastplates, as has been identified in other areas of the Atacama Desert [52, 59–62].

## Results

### Bioarchaeology

**Coastal Archaic Period.** For this Period, we rely on the previous results published [26], which show that 25% (34/136) of the total sample studied (independently of the degree of body completeness) showed traces or markings of trauma compatible with interpersonal violence (S1 Fig). Male individuals (32%, 23/73) were more affected than female individuals (18%, 11/63), a statistically significant difference ($X^2 = 4.01$; $p = 0.045$). The great majority showed only a single trauma, predominantly consisting of fractures. The most affected region of the body was the skull.

*Cranial trauma.* When considering only individuals recovered with the cranium, the percentage of traumas stands at 24% (23/96) and affected males at 32% (18/56). By contrast, the figure for females decreased to 12.5% (5/40). This difference by sex is statistically significant ($X^2 = 4.94$; $p = 0.026$). The traumas, all healed, were distributed among the facial bones and cranial vault (Fig 3). The frontal bone was the most affected. In the vault, most corresponded to slightly depressed fractures. These traumas consist of an oval or circular contour (diameter ranges from 1.8 to 2.5) with a slight depression or sinking of the external table of the bone (0.2 to 0.3 cm), probably due to the advanced process of bone remodeling that most of them show. Exceptionally, there are more extensive fractures that were in the process of healing at the moment of the individual's death. Other traumas correspond to slight erosions or bone abrasions, affecting only the external table, mainly in female individuals.

*Postcranial trauma.* When considering individuals that were preserved complete or only the postcranial skeleton (or remains of it), the percentage with traumas declines to 15% (13/87) (S1 Fig) and both sexes were equally affected. There were 14 traumas (one individual showed a double trauma), and these were concentrated in the upper extremity (n = 9) and thorax (n = 5). The most affected bone was the ulna (n = 5). Thoracic traumas, which includes healed and lethal traumas, affected the ribs (n = 2), vertebra (lethal trauma), and in two cases just the soft tissues of the thorax (lethal wounds).

*Lethal trauma.* Two percent of the total sample of individuals analyzed (3/136) showed lethal trauma. And of the individuals that suffered traumas, 9% (3/34) were lethal, all males (S2 Fig): (a) one individual (MoC/T0, 35–40 years old) had a complete harpoon, including its lithic point and hook, embedded inside the thorax; (b) another individual (MEC/C1, 17–18 years old) had a fragment of a projectile point embedded in the L2 vertebral body (anterior plane);

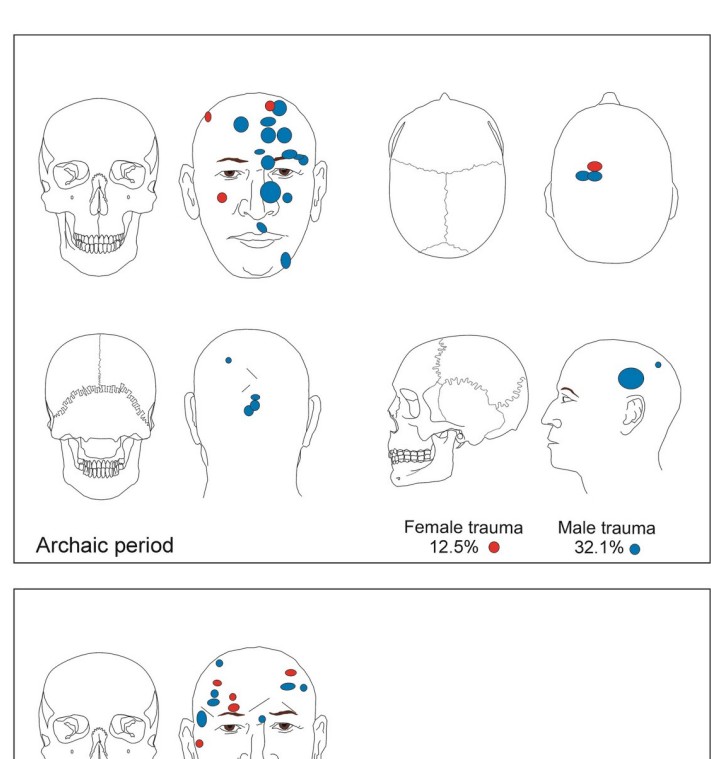

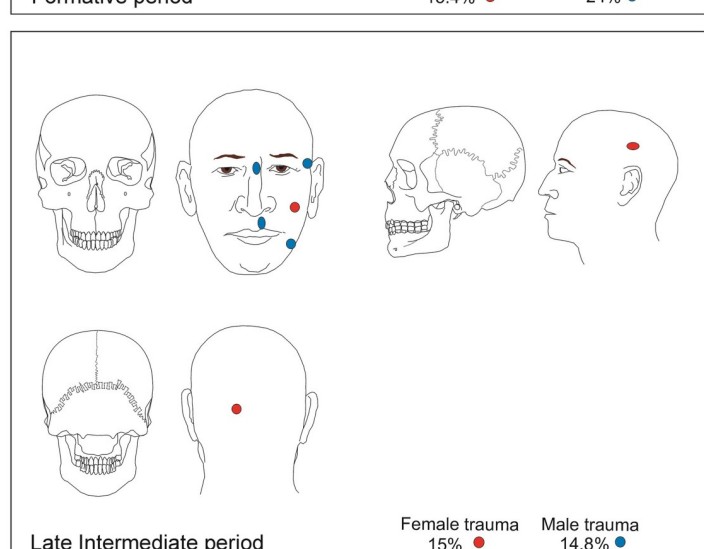

**Fig 3. Distribution of healed cranial traumas.** Archaic, Formative, and Late Intermediate Periods.

(c) another (Mo1/6T18, 30–35 years old), showed four perimortem perforating wounds in the thoracic region, two ventral entry points and two dorsal, whose diameters were compatible with the harpoon diameter measurements [26, 63].

  **Coastal Formative Period.**  Twenty-one percent (17/82) of the sample studied showed traces or markings of trauma compatible with interpersonal violence (S1 Fig). Male individuals

(26%, 10/38) were more affected than females (16%, 7/44), although this difference is not statistically significant ($X^2$ = 2.527 p = 0.112). Most individuals showed only one trauma and, in general, the cranium was the most affected anatomical area.

*Cranial trauma*. The percentage of traumas for those individuals with the cranium preserved was 19% (15/77) and both sexes were represented, with a predominance of males (21.6%, 8/37) over females (17.5%, 7/40) (Fig 3). However, this difference by sex is not statistically significant ($X^2$ = 0.855; p = 0.358).

The healed traumas were exclusively located in the anterior plane of the skull and the frontal bone was the most affected (Fig 3). Most correspond to depression fractures. The traumas are small in extension (diameter ranges from 1.3 to 1.8 cm), oval to circular in shape, and affected only the external table of the cranial vault (as a dent). Only one individual showed a more severe fracture, associated with a linear fracture. On the other hand, five individuals showed lethal traumas that affected mostly the facial region, and the vault hardly at all.

*Postcranial trauma*. Considering individuals that were preserved complete or only the postcranial skeleton (or remains of it), those affected with traumas decreased to 11% (2/19) (S1 Fig), with only males represented (25%; 2/8). One individual had a healed fracture in the right humerus, while another had a lethal trauma at T7, with an embedded lithic point.

*Lethal traumas*. Of the total sample studied for this Period, 7.3% (6/82) showed perimortem traumas. And of the individuals affected, 35.3% (n = 6/17) were consistent with perimortem traumas (S2 Fig), which may have caused their death. In relation to sex, males (n = 3) were likewise as affected as females (n = 3). Lethal traumas affected: (a) a male individual (PlM-7/Ex4/Sn/1), 20–25 years old, shows a depression fracture located on the left parietal eminence (2.8 x 2 cm in diameter) (Fig 4a); (b) a male individual (Ejército 163/C1), 25–30 years old, shows a depression fracture located in the right frontal bone (2.4 x 2 cm in diameter); the internal table shows that the bone fragment was fractured in three parts and remained adhered in situ (Fig 4b); only the cranial vault was preserved and it is not possible to know whether the facial bones were involved; (c) a female individual (PlM-7/T139), 20–25 years old, who shows severe crushing and deformation of the face (Fig 4c) with multiple radiating fractures in the neurocranium. The fractures are beveled and the preservation of soft tissues allowed some of the bone fragments to be kept in situ; (d) a female individual (Quiani-7/T-13), 25–30 years old, had a perimortem fracture on the mandibular condyle and body, both with oblique trajectory and beveled edges; (e) another female individual (Garibaldi-C1), 25–30 years old, showed multiple traumas in lateral planes of the cranial vault; (f) lastly, one individual (Mo-1/6D/T-10, male, 20–25 years old) showed a perimortem fracture in the T7 vertebral arch and T5 and T6 spinous process resulting from the impact of a lithic projectile. The basal segment of the point was embedded in the bone (Fig 5). The bone fracture has a sharp elliptical contour, consistent with the limbus or section of the projectile point.

**Coastal late intermediate period.** Fourteen percent (10/70) of the sample studied showed traces or markings of trauma compatible with interpersonal violence (S1 Fig). Both sexes were equally affected in percentage terms: males 14% (6/43) and females 15% (4/27). All traumas were concentrated in the skull, with the exception of two individuals who also showed postcranial trauma.

*Cranial traumas*. Considering only individuals with the cranium preserved, the percentage of traumas stands at 15% (10/67) and was the same for both sexes (male 15%, 6/40; female 15%, 4/27). Healed traumas (n = 7) (one individual had two traumas) mostly affected the anterior plane of the skull, in particular the facial bones: nasals (n = 1), maxilla (n = 1), mandible (n = 1), zygomatic (n = 1). In addition, minimal trauma was recorded in the cranial vault, posterior plane (occipital), and left lateral (parietal) (Fig 3). The vast majority of these correspond to depression fractures (1.5 x 2 cm in diameter). Four individuals showed lethal traumas.

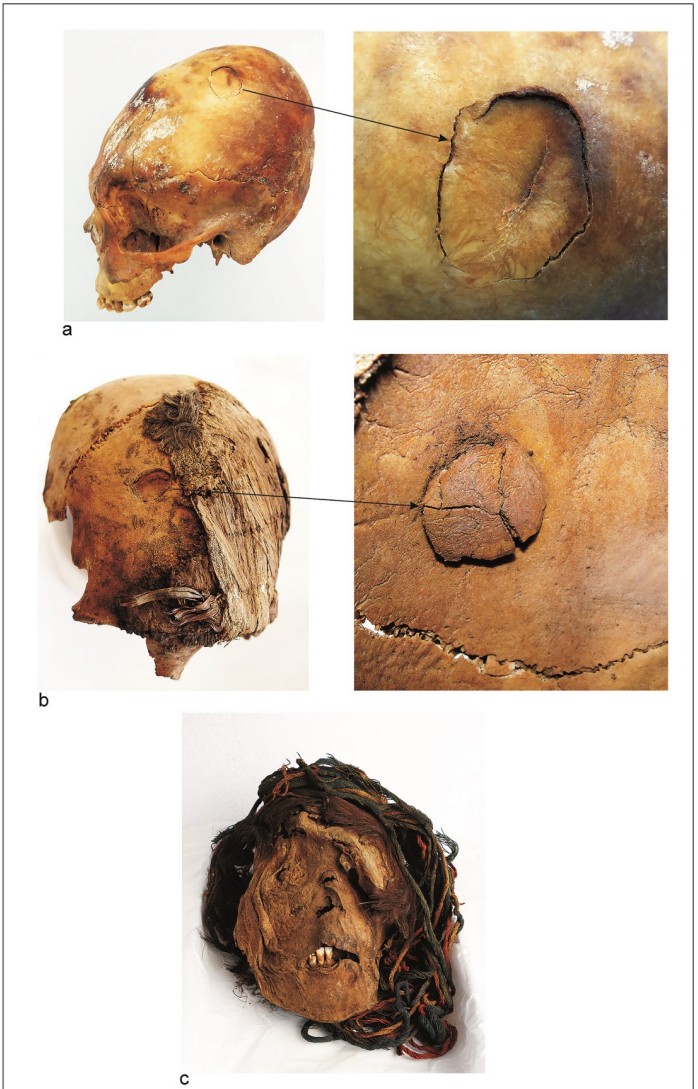

**Fig 4. Perimortem cranium fractures, Formative Period.** (a) Cranium of male individual (25–30 years old) with perimortem fracture in depression (2.3 x 1.7 cm diameters) in the left parietal, probably caused by a lithic ball impact. Site PlM7/Ex4/Sn1, $^{14}C$ 1900 ± 30 BP (1875–1718 cal BP). Beta-539225. (b) Cranium fragment, male individual (25–30 years old) with perimortem fracture in depression (2.4 x 2 cm diameters) in the right frontal, probably caused by a lithic ball impact. Hair remains are preserved and covered by a twining mat. Detail of fracture internal table, the fragment was fractured in three parts. Site Ejército 163/C1, $^{14}C$ 2580 ± 30 BP (2644–2490 cal BP). Beta-491253. (c) Skull of female individual (20–25 years old) with lethal trauma from a high impact blow, probably with a mace. The soft tissues allowed the bone fragments to remain *in situ*, permitting observation of the severe facial deformation. The hair was intentionally cut, and the turban destroyed. Site PlM-7/T139, $^{14}C$ 2260 ± 30 BP (2328–2154 cal BP). Beta-539221.

*Postcranial traumas.* Considering individuals that were preserved complete or only the postcranial skeleton (or remains of it), the percentage affected with trauma decreases to 7% (2/27) (S1 Fig), affecting one male (5.8% n = 17) and one female (10% n = 10). Both correspond to perimortem stop fractures in the forearm (left radius and ulna).

*Lethal trauma.* Of the total sample studied for this Period, 5.7% (4/70) showed perimortem trauma. And of the individuals affected with trauma, 40% (n = 4/10) showed perimortem trauma (S2 Fig). Males (n = 3) were three times more affected than females (n = 1): two

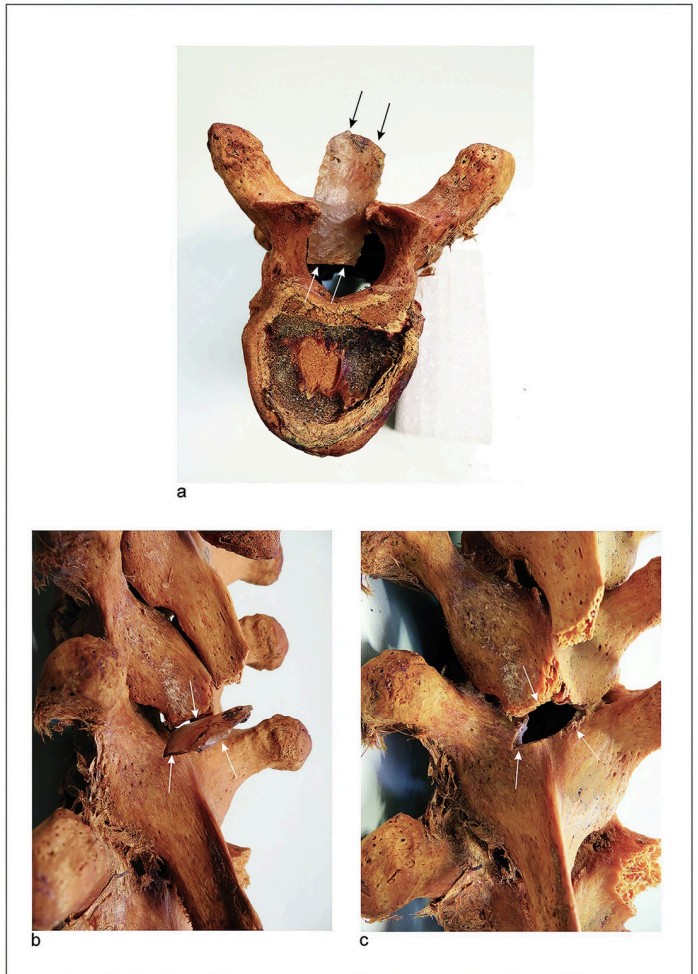

**Fig 5. Male individual (20–25 years old) with lithic point imbedded vertebral arch of T7.** (a) Superior view of T7, the point penetrated the spinal canal and must have severed the spinal cord. (b) Left lateral view. (c) Posterior view, elliptical contour fracture, consistent with the form of the point. The projectile also damaged the spinous process of T6. Probably was thrown using an atlatl. Formative Period, Site Morro 1/6D, Body T10, ca. 2800 BP.

individuals both male (PlM9/T-25, PlM9/T-38), aged 20–25 years old, and one female (PlM4/T-3), 17–18 years old, showed multiple crush fractures in the facial bones, in addition to radiating and beveled fractures in the frontal, parietal and temporal bones; one of them (PlM9/T-38) also had a perimortem fracture in the middle diaphysis of the left radius and ulna; the fourth individual (PlM4/Sn9), male, 20–25 years old showed a more focused perimortem fracture in the body of the left mandible.

## Geoarchaeology

The Sr isotopic signal of the coastal populations from the three chronological periods sampled, corresponding to 37.5% of the total sample studied (108/288), did not vary significantly (Fig 6; Table 2). Ninety-five percent of the individuals (105/108) showed an isotopic signal closer to the signal of marine fauna (both archaeological and modern) than the signature of terrestrial fauna captured at the coast, the oases, the foothills and the altiplano (Fig 1). This marine signal attenuates slightly towards the LIP. And only 5% of the individuals (5/108) showed a non-marine, i.e., non-local, signal.

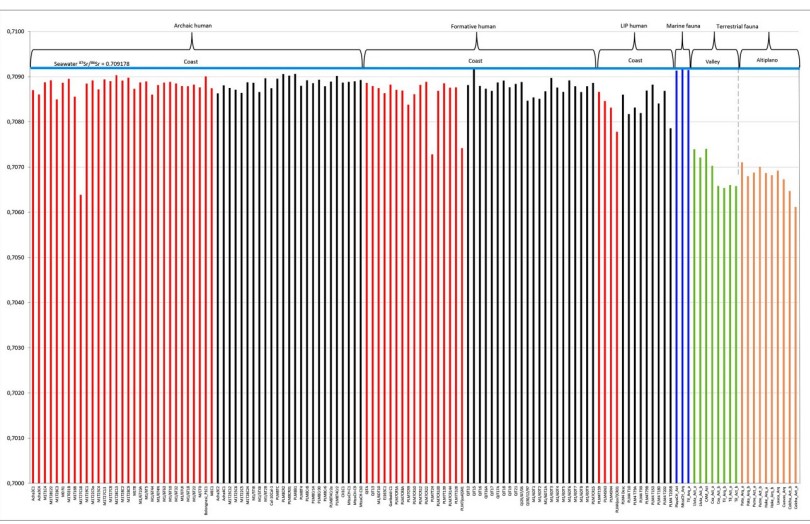

**Fig 6. Graphic of Sr isotopic values for 108 individuals sampled.** Red bar, individuals with trauma; black bar, individuals without trauma; blue bar, marine fauna; green bar, valley terrestrial fauna (coast and oasis); brown bar, foothills and altiplano terrestrial fauna.

When we evaluated the results in terms of timescale, we see that, for the Archaic Period, sampled individuals (n = 56) both with traumas (n = 31) (range $^{87}Sr/^{86}Sr$ = 0.709034–0.708497) and without (n = 25) (range $^{87}Sr/^{86}Sr$ = 0.709061–0.708632). Sr values did not vary between the two groups (Fig 6). Only one male individual showed a non-marine signal ($^{87}Sr/^{86}Sr$ = 0.706385) as well as a healed cranial trauma [26]. For the Formative Period (n = 39), individuals with trauma (n = 17) show a slight dispersion, with values ranging from $^{87}Sr/^{86}Sr$ = 0.708884–0.707281. In contrast, individuals without trauma (n = 22) show a more homogeneous marine signal ($^{87}Sr/^{86}Sr$ = 0.709176–0.708467). For the LIP, although the sample is smaller (n = 13), individuals with trauma (n = 4) ($^{87}Sr/^{86}Sr$ = 0.708665–0.707781) and without (n = 9) ($^{87}Sr/^{86}Sr$ = 0.708823–0.707859) also show a greater dispersion with intermediate values among the marine and valley signals (Fig 6).

## Rock art and geoglyphs

Of the total number of rock art and geoglyph sites reviewed from the coast and low valleys, scenes depicting anthropomorphic motifs in confrontations, brandishing weapons and/or wearing helmets were identified in 204 motifs (149 in engravings and 55 in geoglyphs). The motifs identified in rock engravings are distributed among a total of 14 sites in the valleys of: Lluta (LL-38, LL-98); Azapa (Az-29, Anm-1, Anm-2, Sobraya, Chamarcusiña); Codpa (CoBl-2, La Ladera, Ofa-1, Ofa-2), and Camarones (Huancarane-1, Tal-1, Tal-2). In the case of geoglyphs, these motifs come from 10 sites of Lluta (LL-18, LL-60, LL-89, LL-111, LL-112, LL-113) and Azapa (Az-18, Az-63a, Az-63b, Cerro Sombrero) valleys. Of the total number of motifs identified in rock engravings and geoglyphs, 12 figures (archers facing each other with or without helmets) can be assigned to the Formative Period, and 192 motifs (archers facing each other, anthropomorphs with helmets, anthropomorphs with bolas, and ceremonial knives or tumis) to the LIP (Fig 7).

During the Formative Period, motifs suggesting conflict (n = 12) correspond to scenes of archer confrontation, with cephalic headdresses in rock engravings (sites: Az-29, Anm-1, Anm-2 and Sobraya, from the Azapa valley, located between 5 and 15 km from the coast; and three from La Ladera site located in the Codpa valley at 60 km from the coast) (Fig 7).

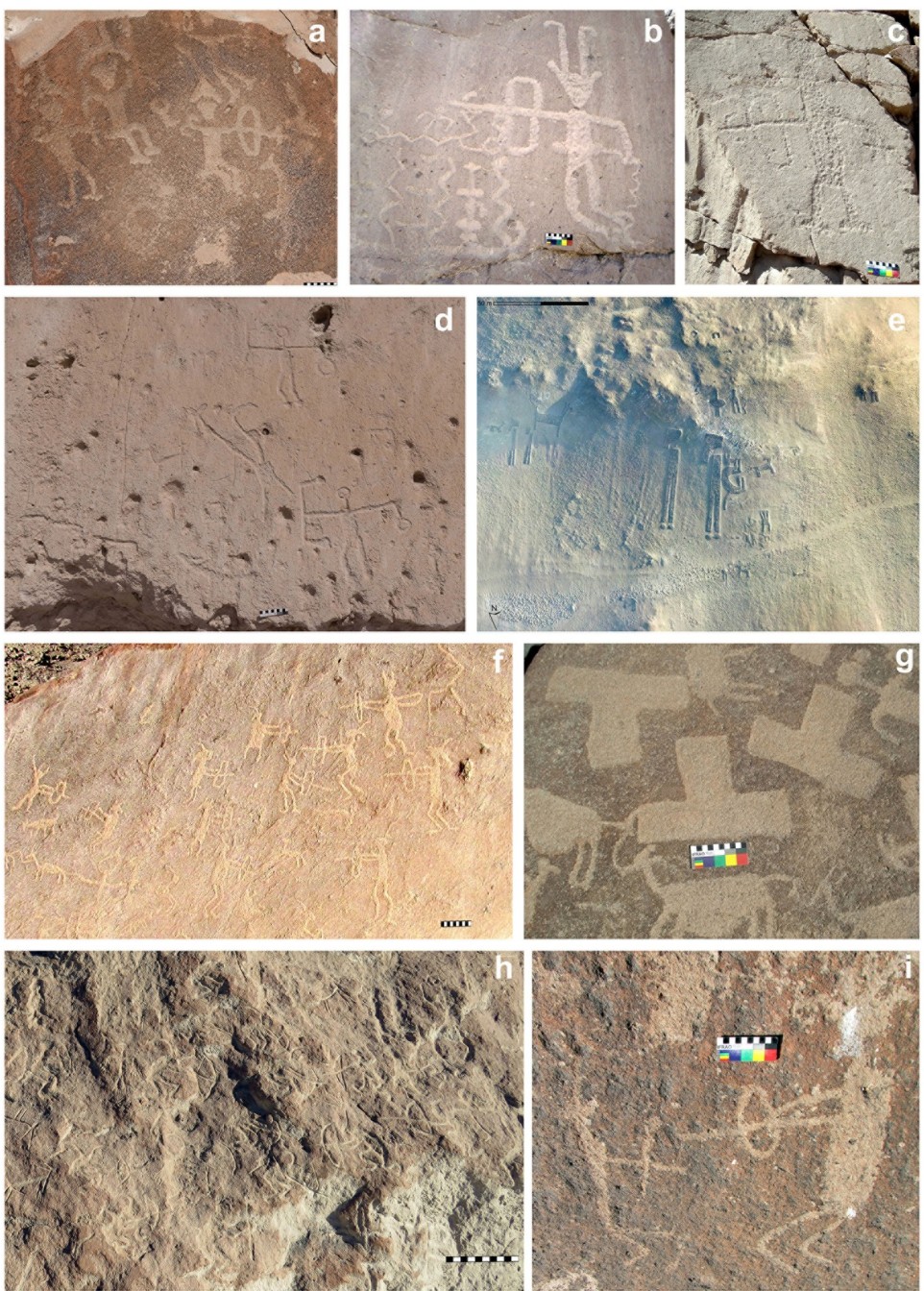

**Fig 7. Motifs in rock art and geoglyphs from the Formative Period (a-c) and Late Intermediate Period (d-j).** (a) Anthropomorphs engraved, facing each other and carrying bows and headdresses, Ánimas-1 site, Azapa valley, photograph by Nick Charlesworth. (b) and (c) Anthropomorphs engraved carrying bows and headdresses, La Ladera site, Codpa valley. (d) Group of engraved anthropomorphs carrying bolas, LL-38 site, Lluta valley, photograph by Nick Charlesworth. (e) Anthropomorphic motifs wearing helmet, geoglyph from LL-111 site, Lluta valley, photogrammetry by Marta Crespo. (f) Archers in confrontation wearing headdresses, engravings from Chamarcusiña site, Azapa valley. (g) Engraved representation of ceremonial knives known as *tumi*, Chamarcusa site, Azapa valley. (h) Scene of dynamic archers wearing headdresses and facing each other, Ofragía-1 engraving site, Codpa valley. (i) Archers in confrontation wearing headresses, Huancarane-1 engraving site, Camarones valley.

During the LIP, the frequency of conflict-related motifs is higher (n = 192), present in both rock engravings (n = 137) and geoglyphs (n = 55). The engraving sites are distributed among the Lluta, Azapa, Codpa, and Camarones valleys, between 17 and 40 km from the coast, at elevations between ca. 500 and 1500 m above sea level (LL-38, LL-43 and LL-98 from the Lluta valley; Ofa-1, Ofa-2, and CoBl-2, from the Codpa valley; Az-49, Chamarcusa and Chamarcusiña from the Azapa valley; Tal-1, Tal-2, and, Huancarane-1, from the Camarones valley). The geoglyphs are located in the Lluta and Azapa valleys, between 4 to 32 km from the coast. The motifs suggesting conflict include confrontation scenes of anthropomorphs in a dynamic attitude wearing headdresses and carrying objects that we interpret as throwing weapons such as bows and darts, anthropomorphs with helmets and with or without objects or weapons in their hands, anthropomorphs with bolas (n = 2), and representations of ceremonial knives known as tumis (Fig 7).

## Trauma, weapons, and modes of fighting or brawls

The traumas are mostly concentrated in the cranium, followed by the upper extremity and thorax. The location of traumas affecting these areas of the body, mainly the cranium, are interpreted as being the result of interpersonal violence [11]. Antemortem cranial traumas, i.e., those showing traces of healing, correspond to depression fractures of the vault. These healed traumas are most frequently located in the anterior plane of the skull, and the frontal bone is the most affected (Fig 3). The weapon that would correlate more closely with this type of trauma is the sling stone. The chronicles [64, 65] points out that the *warak'as* (quechua for sling stone) was one of the characteristic weapons of Andean populations (Fig 8). On the other hand, sling stones have not been recorded for the Archaic Period and it is precisely during this early period that small depression fractures are more common. Around archaic burials, however, it is common to find unmodified spheroidal stones, ranging in size from 3 to 5 cm in diameter (Fig 8). It is striking that these stones, selected for their shape and size, were placed in the fillings of the mortuary contexts. This means that they were not part of the natural soil matrix, which is fine sand. Consequently, we suggest that these stones could have been thrown and caused the cranial trauma. The location of this type of trauma would have implied that individuals were at a moderate distance from one another, because it would have been difficult at a greater distance to target the individual's head, making it easier for them to dodge the impacts of the stones.

It is during the LIP that sling stones are more recurrent, forming part of funerary contexts. The paradox here is that, while sling stones become more frequent towards the LIP, we also see a decrease in depression fractures during this period (Fig 3). This could be explained by the fact that has been recorded for this period, certain types of hats that have been interpreted as "helmets" (Fig 8), and that would have been used in warfare [59]. They were dome-shaped and could protect the cranial vault by means of slats, linked together with bundles of vegetable fiber. The resulting structure was woven with camelid wool dyed in bright colors to form geometric designs (Fig 8). An opening at the front of the "helmet" was designed to expose the face and eyes, allowing the wearer to see. Finally, the helmet was secured to the head with a string tied beneath the chin. Other types of healed injuries located in the facial region, which was not protected by the helmet, correspond to crush fractures, such as nasal, maxillary, and zygomatic fractures. These traumas would not have required artifacts designed as weapons to cause them; rather, they were likely caused by blows delivered with the opponents' bare fists, suggesting body-to-body confrontations in the context of individual fights or brawls or between small groups of individuals.

Fig 9, illustrates the funerary bundle of an adult from the coastal site (PlM-3). It is likely that this individual had a hierarchical role within his group, since he carries a copper axe on

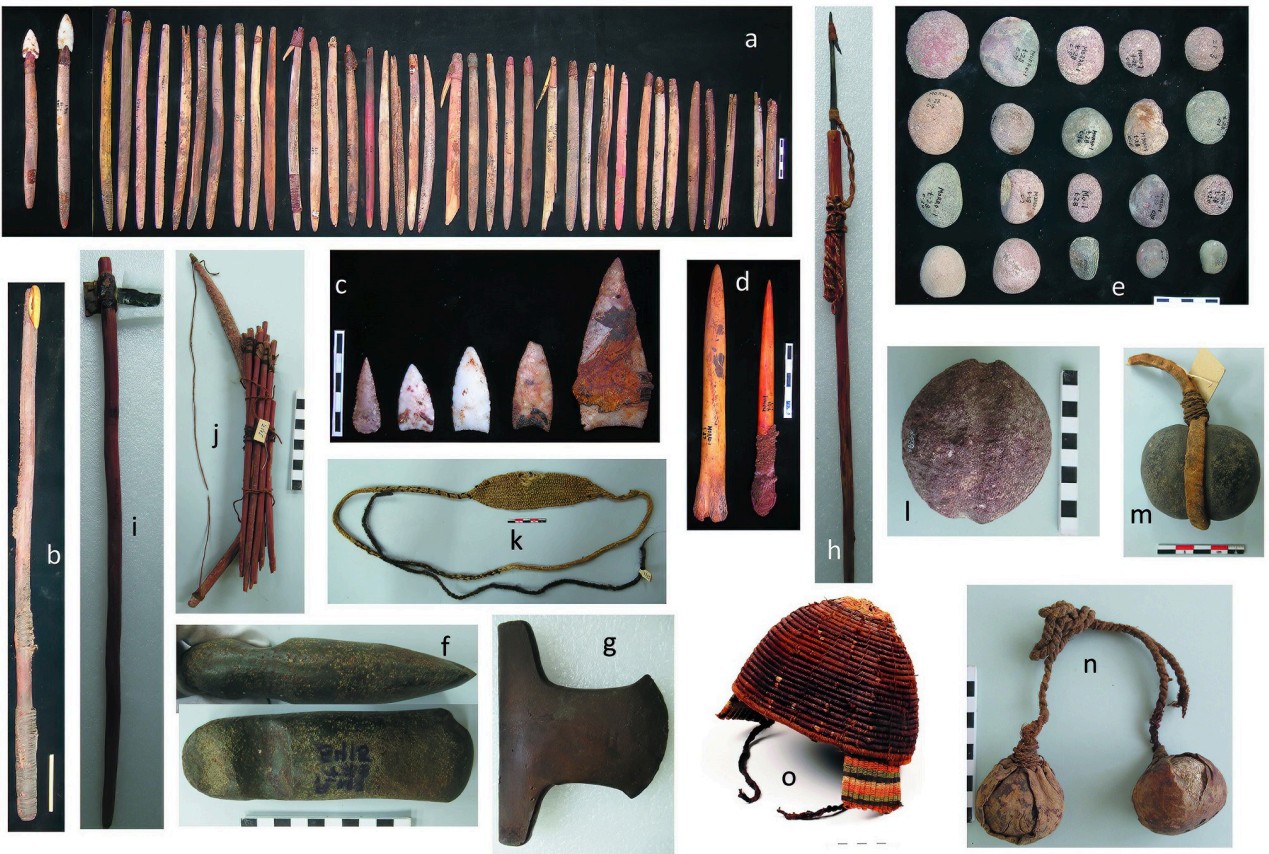

**Fig 8. Weapons and defensive objects.** (a) Harpoons. (b) Spear thrower. (c) Lithic projectiles points. (d) Bone daggers. (e) Small boulders. (f-g) Lithic axes. (h) Cooper harpoon. (i) Lithic ball, (j) Bow and arrow. (k) Sling. (l-m) Sling stone. (n) Boleadora. (o) Helmet.

his left chest, an important symbol of power. In addition, a sling stone had been placed on his right chest as well as spear shafts and a harpoon on the lower back part of the bundle, i.e., the artifacts that, par excellence, coastal groups used as weapons. Thus, it is feasible to infer that this individual had a prominent role in warfare activities.

Healed postcranial traumas mainly affected the upper extremity long bones such as diaphysis of ulna, radius, and humerus. In particular, the ulna diaphysis fracture, known as a parry fracture, is interpreted as an expression of interpersonal violence [33]. When this is the case, the attacked individual instinctively reacts by protecting his face with his hands and forearms, so that the impact of the blows is received directly on the ulna. Although we consider that these traumas reaffirm the idea of body-to-body confrontations, these injuries could have been also caused by accidental falls or in contexts of fights. A few individuals showed coastal fractures, which could have been the result of direct blows from fists to the thoracic region. Although falls cannot be ruled out as an indirect causal mechanism, these may also have occurred in the context of fights or brawls.

In relation to the perimortem traumas, the target was the cranium and thoracic cavity, which could have caused the death of the individuals. Some of them received high impact blows to the skull, destroying the facial region and causing multiple radiated fractures in the neurocranium. To cause these high impact traumas, a blunt weapon must have been used, such as a mace [52]. However, in none of the funerary contexts of the three chronological periods studied were maces identified. By contrast, this weapon has been recorded, although in

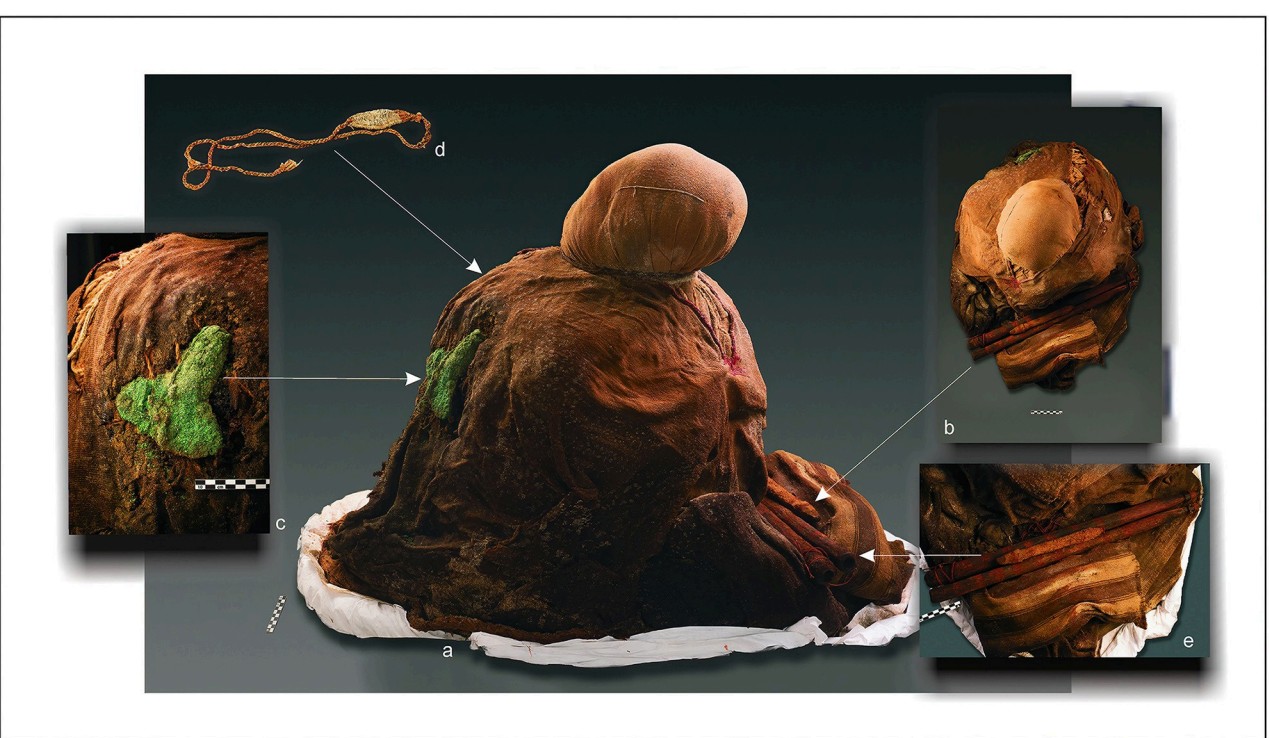

**Fig 9. Funerary bundle of an individual with probable role of "warrior".** (a) Left side view. (b) Top view. (c) A copper axe on the left thorax. (d) A sling on the right thorax. (e) Harpoon and shaft in the posterior and lower of the bundle. LIP Period, PlM-3 Site, Body T308/Bundle-1, $^{14}C$ 630 ± 30 BP (646–588 cal BP). Beta-539227. (Photographs by Nick Charlesworth).

low frequency (n = 6), among the horticulturalists who lived in the adjacent valley, and with whom the coastal populations were neighbors and shared a common cultural matrix. Thus, it is likely that, from the Formative Period, maces were known and used by coastal populations. To cause these lethal traumas with a mace, opponents had to be facing one another in close physical contact, delivering accurate blows that damaged the skull. We have not identified this type of lethal trauma in the Archaic Period (two individuals presented probable high impact perimortem fractures, but were not considered positive cases, due to a lack of solid diagnostic evidence, e.g., preservation of soft tissues or hinged bone fragments). By contrast, we have identified them from the Formative Period onward, and this type of confrontation is maintained during the LIP, although in general they are of low frequency.

Other weapons that show a positive correlation with lethal trauma corresponds to marine hunting weapons such as harpoons, and terrestrial hunting weapons such as atlatls (Fig 8). These weapons, especially the harpoon, are recurrent in the funerary contexts of the three chronological periods. But unlike for maces, interpreting these weapons as being intended to cause damage to other humans is fraught with ambiguity because of their inherent functions in marine and terrestrial hunting. However, given that lithic points were recorded embedded in the bones and soft tissue of some individuals from the Archaic and Formative Periods, including one case in which a complete harpoon head was found inside the thoracic cavity, we suggest that these hunting weapons could also have been used as lethal weapons. For the Formative Period, we identified one individual that had the basal fragment of a quartz lithic point embedded in the neural arch of T7, which fractured both the tip and bone, penetrating the interior of the spinal canal and severing the spinal cord (Fig 5). The impact of the point also

damaged the spinous process of T6 and T5. The absence of any response in the bone tissue suggests that his death occurred shortly following impact. During the LIP, although harpoons continued to be widely used, in addition to the bow and arrow (Fig 8), we did not record any individual with a projectile point impact on the body. With this type of weapon, we again see the paradox that, while hunting weapons such as the bow and arrow are more frequent in the funerary contexts of the LIP, this was not supported by the bioarchaeological evidence. These results are consistent with those reported for LIP populations of the Camarones river mouth, where there were no reported cases of projectile points embedded in the bodies, despite the presence of bows and arrows in the burial contexts. It is likely they were more protected (using helmets and other attire such as tightly woven shirts) against possible violent events. In all three chronological periods, these artifacts are frequently placed beside the bodies as part of the mortuary contexts (Fig 9) [26, 37, 40, 41]. It suggests that coastal populations assigned a high value to these types of hunting weapons, particularly the harpoon, and the bow and arrow during the LIP.

This study supports the view that head trauma is the best predictor of interpersonal violence. The traumas affecting the neurocranium, depending on the intensity and location of the blows, may have caused serious damage such as: loss of consciousness, drowsiness, balance difficulties, seizures, neck stiffness, and nausea, some of which would have caused periods of acute disability for individuals exposed to such injuries [66]. These traumas could also have left individuals with lifelong sequelae [66], resulting in severe limitations that would have required care and support from close members of the social group. In addition, it is likely that even a low intensity trauma affecting only the soft tissues would have caused the richly irrigated lip mucosa to generate abundant bleeding, ecchymosis, swelling, and deformation of the face [11].

When throwing devices were used as weapons of attack, such as the atlatl and bow, or when harpoons and knives were used as daggers, the target was usually the postcranium, particularly the thoracic region. It has been reported that among patients of the 19th century American Indian wars and those of modern warfare in New Guinea, projectile wounds affected the thorax with greater predilection (33.1% and 36.4% respectively) compared to other regions of the body [54]. This is the most vulnerable part of the human body to receive the impact of sharp weapons, causing highly lethal perforating wounds, given the location of vital organs and the little bone protection they have compared to the cranial vault. Thus, when collections of skulls alone are studied, lethal traumas affecting the postcranial skeleton will be entirely invisible, partially biasing the interpretation of past violence.

**Trauma versus sex and age of individuals.** When we analyzed the trajectory of violence in relation to the sex of the individuals, we observed that male violence tends to decrease over time (although this difference is not statistically significant). By contrast, female individuals with traumatic injuries due to interpersonal violence remain clearly invariant over time. Some individuals affected with lethal trauma correspond to young adults, with males predominating over females. However, we also identified one female in the Formative Period with a perimortem mandibular fracture, and three females (one in the Formative and two in the LIP) who died from high impact blows to the head. These data, although scarce, raise the question of whether lethal violence against women increased in the late periods. Another expression of violence that remains relatively invariant is the age of the affected individuals, where trauma due to interpersonal violence occurs primarily against individuals between the ages of 20 and 35. These results are consistent with ethnographic evidence [10]. Studies of skeletal series also show a predominance of male individuals with traces of trauma from interpersonal violence [44, 50, 51, 67].

## Discussion

When contrasting the results of our different lines of evidence with the proposed hypotheses, we verify that in these fishing, hunting and gathering societies that inhabited the coast of the Atacama Desert, violence and conflict were present and rather constant during the 10,000 years. When we evaluate the dynamics of violence during the Archaic Period, which spans more than 6000 years, we see that violence was invariant from the Early Archaic (10,000 cal BP) to the Late Archaic (4000 cal BP) [26]. This suggests that population increase during the Late Archaic did not trigger violence. When we expand the timescale and analyze the bodily traces of violence in Formative populations (3000–1500 BP), we again observe that violence remains invariant with respect to the Archaic Period. The novelty observed during the Formative Period is that the lethality of violence increases substantially compared to the Archaic Period. This same phenomenon was observed in Formative horticulturalist populations of the Azapa Valley, where 50% of the traumas suffered by individuals were lethal [52]. These results are consistent with a comprehensive study of warfare and violence in the Andean region [44], which show that, although warfare and violence varied across time and space, the first major wave of widespread violence occurred during the Formative Period [44].

During the Archaic Period, coastal populations lived for thousands of years in relative isolation, without close neighbors, since the valleys and quebradas, were not permanently occupied. Despite living their daily lives next to the Pacific Ocean that provided the most important resources to the subsistence, these coastal populations moved freely throughout the territory via routes that connected the coast with the high puna. By contrast, during the Formative Period, other groups from the south central Andes area probably began to interact more frequently with the coastal populations, which generated substantial changes in the social, cultural, and economic dynamics of these millennia-old maritime traditional populations.

The greatest impact for them was the introduction of people with a new subsistence strategy: horticulture, which caused a diversification of the diet maintained for millennia based mainly on marine resources. As people began to settle in the valleys, giving way to villages and cultivated fields [28, 68] neighbors emerged within 10–20 km of the coast. These people differentiated themselves from fishermen by forging their identities linked to plant cultivation. This economic differentiation between horticulturists and fishermen, however, did not prevent them from sharing certain cultural traits (e.g., funerary patterns), but the coastal population continued their traditional way of life of fishing, hunting and gathering marine resources.

A similar scenario was observed during the LIP, when social complexity increased. According to ethnohistoric information [30, 31] and archaeological interpretations [69], fishing and farming societies were organized under simple chiefdoms, with leaders who maintained some territorial and political control, interdigitated within the valleys and along the coast. These local leaders were subordinate to a larger authority located inland, as part of supralocal political groupings [32]. A common constant among them was "the possession of beaches, coves and lagoons belonging to each village" [32]. The "possession" of beaches would suggest a sort of "private property" over the fishing resources of these enclaves, which could have triggered situations of conflict and tension among the groups for access to, and control over, these coveted resources (e.g., seabird guano, dried fish, algae, and ritual objects such as starfish and shells).

As for the line of evidence provided by strontium (Sr) isotopes, of the total number of individuals sampled (n = 108), from all chronological periods, 95% (n = 103) have high Sr values, ranging from $^{87}Sr/^{86}Sr$ 0.709176 to 0.708172 consistent with a marine signal. Only five individuals (5%), from the three chronological periods, show Sr values closer to a valleys signal (ranging from $^{87}Sr/^{86}Sr$ 0.707859 to 0.706385), demonstrating that over millennia, non-local individuals buried on the coast of Arica was exiguous. On the contrary, these Sr isotopic values

demonstrate that these millennia-old fishing, hunting, and gathering populations were born, and lived their daily lives. These result are consistent with $\delta^{13}C$ and $\delta^{15}N$ isotope values from other studies [70–74] that show principally a marine-based diet for coastal populations in the extreme north of Chile.

When we compare the chemical signature of Sr between individuals with trauma and those without, we see that they do not differ from one another. These results suggest that violence would have occurred mostly in a context of tension and conflict between individuals or groups of local individuals. This is based on the minimal presence of individuals with traumas (n = 4) that show a non-marine signal, i.e., they were not born on the coast, but were buried in this environment.

In relation to the timescale, we did not identify any foreign individuals in the Early and Middle Archaic Periods. In the Late Archaic, one individual (M1T27C18, male) was identified with a non-marine Sr chemical signature, showing that it came from another geological local-ity. Given that the value of the chemical signature was slightly lower than the range we have for the coastal valleys in the extreme north of Chile, and given the values presented for the Moque-gua valley [75], we suggest this individual may have come from the valleys in the extreme south of Peru [26]. It is surprising that, from a range of 6000 years, we have only identified one foreign individual, which would suggest a certain degree of inbreeding within these archaic populations, as has been suggested in previous work [76].

For the Formative Period, only two individuals (PlM7/T24, PlM7/Sr1, both male), have a non-marine Sr chemical signature; their values are concordant with the Sr chemical signature of terrestrial fauna (wild rodents) from adjacent valleys. This would indicate that these two individuals were born in one of the valleys of Arica and, at some stage in their lives, moved to live on the coast. The occurrence of these two non-coastal individuals suggests the possibility of hostile social interactions between the fishers and their close neighbors, the Azapa Valley horticulturists, who lived 5–17 km from the coast. Even more so, given that the horticulturalist populations were also exposed to high levels of violence with high lethality [52]. The fact that the chemical signal of the Sr for some of the horticulturists shows an intermediate signal between the valley and the marine demonstrates the dynamics and interaction between both groups and between both environments for reasons of complementarity, economic relations, social ties, marriages, among others, which would not have been exempt from episodes of hos-tility. Moreover, some individuals were buried naked, without grave goods, in atypical and forced positions (Fig 10a and 10b) [77]. This suggests an attempt at mutilation to which these individuals could have been exposed. These same behaviors have been verified in some Forma-tive Period individuals of neighboring horticulturalists from the Azapa Valley [42, 52, 77], so it could have been a practice during this Period.

During the LIP, we see a somewhat different picture. Although the chemical signature of Sr during this Period continues to be predominantly marine, individuals show slightly lower values than the Archaic and Formative groups. [70]. However, two individuals (PlM4Ex7/Cr01, male with trauma, and PlM4/T205B, female without trauma), exhibited intermediate values between the marine and valley signals. The values suggest that these individuals could have moved between the coast and adjacent valleys to exchange products, and that they would have had access to plant products, incorporating these into their diet, albeit always in low amounts [68].

## Rock art and geoglyphs

Rock art and geoglyphs sites at the northern Chile were evaluated to find out whether any of the images depict scenes of combat or battle with human beings facing one other, carrying weapons or having warrior attributes. During the Formative Period, scenes of archer

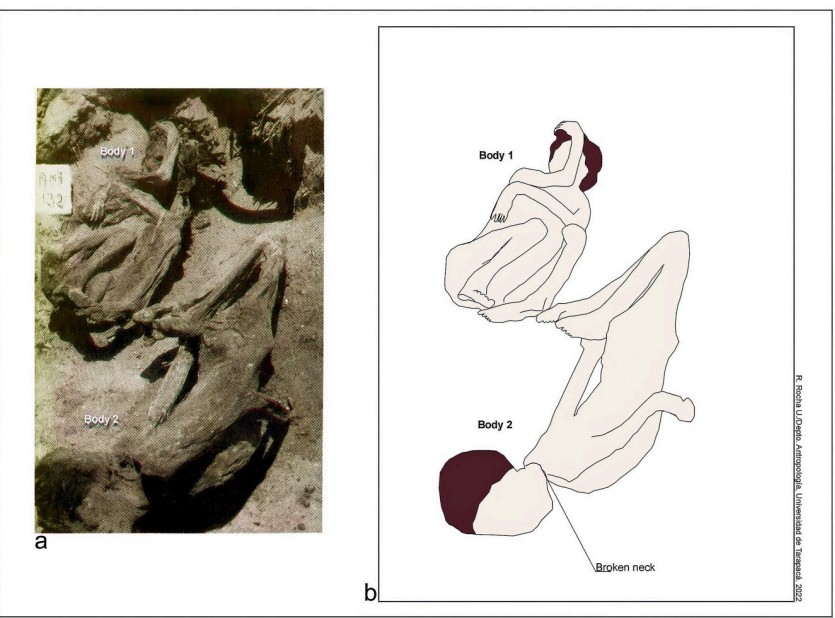

**Fig 10.** a, b. Naked individuals, without grave goods. Body 1, in a seated position, hyperflexed, its right arm covering its face, as if trying to protect himself. Body 2, shows a clearly forced position with the trunk and thighs arched in hyperextension, the knees hyperflexed, with the feet on the buttocks and hands towards the back; the head was detached from the neck. Formative Period (PlM7 Site, 2500 BP).

confrontation, with cephalic headdresses, at sites located in the Azapa Valley suggest the symbolic representation of conflicts demarcating a new economic space: the horticultural. It is interesting to note that two of the sites from this Period (Anm-1, Anm-2) are located near a freshwater spring. It is possible that competition for this type of space and resources generated social tension and conflict, as evidenced in the bioarchaeological record. Further south, some rock paintings in the Confluencia style from the Formative Period in the Loa River include anthropomorphic figures facing each other, in dynamic actions, wearing headdresses and carrying throwing weapons in their hands, interpreted as spear throwers and darts. Our findings of individuals with projectile points embedded in their bodies demonstrate that these weapons were indeed used to inflict damage on an opponent.

During the LIP, rock engravings have been documented with scenes that clearly combat [61, 62, 78]. The greater number of motifs alluding to conflict compared to the Formative Period is probably due to the greater number of rock art sites from the LIP. This is a consequence of a growing increase in this cultural practice, both along the coast and inland. There is also a greater variety of motifs alluding to conflict than in the previous period. Many human figures wear headdresses, which shows the marking of social differences, suggesting the representation of different groups. Also, the representations of tumis suggest the notion of prestige. Many of the LIP motifs identified are associated with representations of llama caravans, suggesting that exchange and conflict were closely linked [56]. This has been proposed for Santa Barbara style at the upper Loa River where are petroglyphs with scenes of caravanners and supposed "warriors" carrying helmets and pectorals [56].

Several researchers [21, 59–62, 78, 79] have discussed whether these rock art images from northern Chile represent scenes of warrior confrontation in actual battles or historical events; or, conversely, whether they are representations of rites and Andean ritual battles, such as *tinkus* [61], which are part of the Andean cultural matrix [21]. Given the bioarchaeological

evidence, we interpret these scenes as symbolic representations of historical events that express situations of social tension and conflict from the Formative to Late Periods. These scenes seem to have been part of the visual discourses in the context of conflicts during the prehistory of northern Chile [56, 59, 62, 80].

**What does the settlement pattern of these coastal societies reveal in relation to the practice of violence?.** While bioarchaeology reveals traumas due to interpersonal violence, as well as the weapons required to cause such traumas, the settlement pattern of coastal populations does not show defensive features. This is to be expected, given that in general, such complex architecture is not characteristic of hunter-gatherer societies. Fisher's camps on the south central Andes coast were, in constructive terms, very simple [36, 81, 82].

Within our study zone, the geomorphology of the northern coast of Arica is substantially different from that of its southern coast. The former's coastal mountain range is submerged in the sea, and the absence of cliffs gives way to a low relief landscape from which an extensive alluvial terrace stretches northward from the mouth of the San José River until it connects with the mouth of the Lluta River (Fig 2). Here the burial sites are small, comprising four to twelve individuals in a dispersed pattern (Terraza Chinchorro, Lluta), unlike what occurs on the southern coast where the sites have a high concentration of bodies.

Another difference is that domestic sites on the north coast correspond to small camps, which contrast with the large shell middens of the south coast that can reach heights of four to five meters of accumulated debris derived from daily life. The open and exposed littoral enclaves of the northern coast, with no natural protection, contain a diversity of resources concentrate at the mouth of the Azapa and Lluta rivers sach us: springs with drinking water, aquatic plants (e.g., reed), avifauna, fish and bivalves inhabitnts of the sandy beaches, very attractive to hunter-gatherers and fishermen. The reason why it was less inhabited than the southern coast may be due to the lack of natural protection against possible ambushes or surprise attacks. Thus, the greater predilection for inhabiting the southern coast was, beside the abundance and diversity of marine resources, the characteristics of its abrupt geomorphology that provided better natural protection.

During the Formative Period, when lethal violence increased and the possibility of dying in a violent encounter was three times greater than during the Archaic Period, populations did not live more on the northern coast. Another peculiarity is that, during this Period, we have identified five individuals buried in isolation in the pampas adjacent to the coast and with no funerary bundles or offerings. Although these individuals do not show traces of violence on their skeletons, we believe that their isolated burials outside the cemeteries could be explained by a violent death far from their settlements, with injuries that only affected the soft tissues. The architecture of the coastal settlements show no defensive features for the three chronological periods studied, which contrants with the highlands of northern Chile, as well as throughout the entire south central Andes area during the LIP, where defensive settlement known as *pukaras* were very common.

**Why was there violence in these fishing, hunting, and gathering societies living along the Pacific Ocean?.** The mouths of the Azapa and Lluta valleys were very attractive ecosystems for human settlements with different levels of social and economic organization because of fresh water resources and an abundant diversity of coastal marine bioproductivity. These conditions of environmental circumscription could be one of the external factors explaining why these societies engaged in endemic violence for several millennia.

Social organization remain rather egalitarian during the hunting and gatheric ecommic system of the Archaic phases, and became more complex since the Formative Period up to the LIP, with levels of social stratification. Social groups maintained small-scale structures with limited population sizes and without any centralized political control. Documents from the

16th and 17th centuries [32] indicate that the fishing settlements of Moquegua (Ilo), Tacna, Arica, and even Atacama, were governed by their own coastal ethnic leaders, although subordinate to the agricultural leaders (caciques) [32]. Both groups–fishers and farmers–maintained not only relations of complementarity, but also tension and conflict.

In the most comprehensive study on warfare and violence for the Andean region, base on settlement patterns and cranial trauma, two waves of great violence stand out [44]: the Formative period and LIP. In this context, the authors realized that during the LIP period the highest frequencies of cranial trauma occurred among societies in territories with little political centralization, such as Andahuaylas [44] and San Pedro de Atacama in the high land of northern Chile [50]. Similiarly, the coastal populations of northern Chile from the Archaic to the LIP that never lived under complex stratified social structures with a centralized political organization, maintained high levels of interpersonal violence. Based on ethnographic data, Keeley (10) compares the frequency of warfare and its casualties between small-scale (hunter-gatherer and tribal groups) and complex societies (chiefdoms and states with high social stratification), finding that the former practiced warfare more frequently. These data show that warfare was more frequent among small-scale societies [10, 17]. Moreover, ethnographic studies [10, 17] point out that, in pre-state, decentralized, small scale societies, violence occcurr at lower scale, characterized by individual or group skirmishes, brawls or assaults, caused by revenge, witchcraft, and curses. Among these issues, revenge led to a spiral of violence that was difficult to break. These triggers of violence are difficult to contrast empirically with data from extinct populations, and even more so in the absence of direct ethnographic data.

## Conclusions

The results of this study show that the social life of the fishing, hunting, and gathering populations of the Atacama Desert in the extreme north of Chile was not entirely socialy harmonious. On the contrary, they lived immersed in millennia-old endemic violence. Traces of lethal and non-lethal violence on bones and soft tisues, the use of weapons, and rock art representations, support the notion that populations faced conflicts and tensions that, some times, were resolved by violent means. The chemical signature of Sr for individuals sampled suggests that these fights and brawls were generated in the context of local groups. From the Formative Period onward, the internal conflicts of the fishermen could have evolved towards the horticulturists groups inhabitans of neighboring valleys with whom they interacted regularly.

Contrary to expectations, Archaic populations were more exposed to violent contexts than later ones. This decrease in trauma in the later populations, however, is not statistically significative. A peculiar feature is that, while non-lethal violence slightly decrease over time, by contrast, lethal violence increased dramatically from the Formative Period onward and remained unchanged in the LIP. One constant among these coastal groups is that they remained as small-scale societies with an absence of highly centralized political power for the 10,000 years of their existence, which would explain the high rates of violence. In addition, competition for resources in extreme environments such as the Atacama Desert could be another explanatory factor and, finally, this long-term trend of violence is not explained by fighting against foreign groups, according to isotopic evidence.

## Supporting information

**S1 Table. List of the 288 individuals analyzed for this study from the coast of Arica, northern Chile.**
(XLSX)

**S1 Fig. Percentage and relative frequency of individuals with trauma.** Archaic, Formative, and Late Intermediate Periods: gray bar, considers the total N of the sample regardless of the degree of completeness.
(JPG)

**S2 Fig. Percentage and relative frequency of individuals with lethal and non-lethal trauma.** Archaic, Formative, and Late Intermediate Periods.
(JPG)

## Acknowledgments

To Evelyn Coleman for her assistance in strontium sample preparation and analysis in the Isotope Geochemistry Laboratory, Department of Geological Sciences, University of North Carolina at Chapel Hill. To Anita Flores for her field and Bioarchaeology Laboratory assistance at the Universidad de Tarapacá; Eduardo Palma from Pontificia Universidad Católica de Chile, for providing modern rodent samples from the highlands of Putre and Colchane; Raul Rocha for drafting maps and figures, except Fig 2 by Patricia Henriquez, Fig 7e by Marta Crespo, Figs 7a, 7d and 9 by Nick Charlesworth. The Fig 10a, is reprinted from Chungara Revista de Antropología Chilena 19, 1983, Fig 1, page 141, authored by Patricia Soto-Heim; and with permission from Héctor González, Editor of the Journal. Permissions for isotopes and radiocarbon analyses outside of the country were granted by the Consejo de Monumentos Nacionales (Diario Oficial República de Chile #42.449 and #42.807; CMN #846 and #661).

## Author Contributions

**Conceptualization:** Vivien G. Standen, Calogero M. Santoro, Daniela Valenzuela, Bernardo Arriaza, John Verano, Pablo A. Marquet.

**Data curation:** Vivien G. Standen, Daniela Valenzuela, Bernardo Arriaza, John Verano, Susana Monsalve, Drew Coleman.

**Formal analysis:** Vivien G. Standen, Calogero M. Santoro, Daniela Valenzuela, Bernardo Arriaza, Susana Monsalve, Drew Coleman.

**Funding acquisition:** Vivien G. Standen, Calogero M. Santoro, Daniela Valenzuela, Bernardo Arriaza.

**Investigation:** Vivien G. Standen, Calogero M. Santoro, Daniela Valenzuela, Bernardo Arriaza, John Verano, Susana Monsalve, Drew Coleman, Pablo A. Marquet.

**Methodology:** Vivien G. Standen, Calogero M. Santoro, Daniela Valenzuela, Bernardo Arriaza, John Verano, Susana Monsalve, Drew Coleman, Pablo A. Marquet.

**Project administration:** Vivien G. Standen.

**Resources:** Vivien G. Standen.

**Writing – original draft:** Vivien G. Standen.

**Writing – review & editing:** Vivien G. Standen, Calogero M. Santoro, Daniela Valenzuela, Bernardo Arriaza, John Verano, Susana Monsalve, Drew Coleman, Pablo A. Marquet.

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
