## [Decision Letter · Decision Letter 0]

11 Nov 2022

PONE-D-22-28217Violence in fishing, hunting, and gathering societies of the Atacama Desert coast: a long-term evolutionary perspective (10,000 BP - AD 1436).PLOS ONE

Dear Dr. Standen

Thank you for submitting your manuscript to PLOS ONE. After careful consideration, we feel that it has merit but does not fully meet PLOS ONE’s publication criteria as it currently stands. Therefore, we invite you to submit a revised version of the manuscript that addresses the points raised during the review process. I agree with the issues raised by both  reviewer which are major and should be addressed.  Reviewer 1 points outaspects regarding the methods, tand the exclusion of subadults. And also has issues with the interpretation and some of the phrases and language that is used.Reviewer 2recommend reorientation of the paper around clear research questions that can be addressed with the data that is used in the paper. In particular there is no clear rationale as to why theere is an underlying assumptopn of evolution and more so of "social evolution" which is in actuality a problematic approach. Reviewer 2 suggests either removal of the untestable ‘evolutionary’ framework, or simple acknowledgement that the debate exists but that the current work cannot directly address those larger evolutionary questions and will focus on whether interpersonal violence corresponds with specific cultural variants over 10ky of prehistory. There are several additional detailed comments and all should be addressed. I believe that the outcome will be productive and  that the comments are useful and constructive although the process will involve a reorientation of the manuscript (and not just some more minor changes to the existing text) Please  submit your revised manuscript by December 26, 2022. If you will need more time than this to complete your revisions, please reply to this message or contact the journal office at plosone@plos.org. Please include the following items when submitting your revised manuscript:A rebuttal letter that responds to each point raised by the academic editor and reviewer(s). You should upload this letter as a separate file labeled 'Response to Reviewers'.A marked-up copy of your manuscript that highlights changes made to the original version. You should upload this as a separate file labeled 'Revised Manuscript with Track Changes'.An unmarked version of your revised paper without tracked changes. You should upload this as a separate file labeled 'Manuscript'.

We look forward to receiving your revised manuscript.

Kind regards,

Ron Pinhasi

Academic Editor

PLOS ONE

2. In your manuscript, please provide additional information regarding the specimens used in your study. Ensure that you have reported specimen numbers and complete repository information, including museum name and geographic location.

For more information on PLOS ONE's requirements for paleontology and archaeology research, see https://journals.plos.org/plosone/s/submission-guidelines#loc-paleontology-and-archaeology-research.

3. We noted in your submission details that a portion of your manuscript may have been presented or published elsewhere. [Data on violence among Archaic period Chinchorro hunters and fishermen from the far northern coast of Chile were published in American Journal Physical Anthropology, Standen et al. 2020, doi: 10.1002/ajpa/.24009. The novelty of this new Manuscript is that it assesses the long-term trajectory of violence in societies that inhabited the coast of the Atacama Desert in northern Chile from the Archaic Period to the pre-European Contact Period, i.e. for 10,000 years.] Please clarify whether this publication was peer-reviewed and formally published. If this work was previously peer-reviewed and published, in the cover letter please provide the reason that this work does not constitute dual publication and should be included in the current manuscript.

Reviewers' comments:

Reviewer's Responses to Questions

**Comments to the Author**

1. Is the manuscript technically sound, and do the data support the conclusions?

Reviewer #1: Partly

Reviewer #2: Partly

2. Has the statistical analysis been performed appropriately and rigorously? 

Reviewer #1: Yes

Reviewer #2: Yes

3. Have the authors made all data underlying the findings in their manuscript fully available?

Reviewer #1: Yes

Reviewer #2: Yes

4. Is the manuscript presented in an intelligible fashion and written in standard English?

Reviewer #1: Yes

Reviewer #2: Yes

5. Review Comments to the Author

Reviewer #1: The context of the study and the questions being posed are well set out, and place in a wider debate concerning the extent of, and variation in, violence in hunter-gatherer societies more generally. In this sense, the paper makes an important contribution.

However, there are some issues regarding the methods employed to address prevalence given the varying state of in/completeness of the skeletons in the sample that needs to be addressed in more detail, even if only in SI. Similarly, there is insufficient information on how the Sr isotope analysis was undertaken.

The interpretation of ca. 95% of the human Sr isotope values as being essentially ‘marine’ needs greater nuance, as none are actually consistent with what is expected from a 100% marine signal, but instead show varying terrestrial input: while this may be generally low (and so 'predominantly marine' might be justified) this is not clear from the information provided, which does not include sample locations for the terrestrial faunal baseline (how close to the coast are they?).

While the paper is generally clear and well written, the language used on occasion is overly dramatic and unwarranted based on the evidence alone (e.g., reference to ‘annihilation’ and ‘slaughter’). Some of the images of soft tissue may be seen as problematic for the indigenous (or those claiming descent) communities of the region (e.g., because of their potential sensitivity, Society for American Archaeology journals would not accept these images, or indeed any images of human remains or funerary objects, without express written consent from representatives of the most appropriate source communities. While this may not be the policy of PLoS ONE, the authors and editors should be aware of this potential issue).

4-92/ Confusing to move from (cal) BP to AD in the cultural periodisation. Suggest one or the other.

6-135/ “evidence of interpersonal violence trauma” > grammar

Why focus on adults only? To what extent was violence also present in subadults? See p 13 where cranial depressed fractures are said to be shallow due to the “advanced process of bone remodeling that most of them show”, implying that they were inflicted earlier in life. All of the traumas for this group show healing, but would this also be the case if older subadults were included? What is the relationship between adult age and their expression?

10-210/ “the presence of fractures in butterfly-shaped was decisive” > grammar

While strontium isotope analysis may fall within the broad remit of ‘geoarchaeology’, the fit is perhaps not the best. Suggest 'biogeochemistry' (not a required revision, a suggestion only).

The Methods section on Sr is inadequate. There is no information on sample pretreatment or analysis, including whether measurements were made using TIMS or MC-ICP-MS, which impacts on their precision.

13-272/ “(independently of the degree of body completeness)”

Explain further how complete vs. incomplete skeletons were analysed, as this is crucial to providing population estimates of trauma prevalence. This is evident from the different prevalence figures provided for only those individuals ‘recovered with the cranium’ [complete?] which present the majority of traumas relateable to interpersonal violence. Ditto for the later periods. Usually it is stated that a certain proportion of the element in question had to be present for it to ‘count’. Should also address how the 20% of individuals (which is more than a ‘few’, p 6) with soft tissues were treated, given that – depending on the degree of soft tissue preservation – the skeleton would not be as visible for macroscopic observation, e.g., of embedded projectile points/fragments.

13-303/ a minor point, but the age of individual MEC/C1 is given as 16-17 years, whereas the Methods section states that adults in the study are defined as >17 years.

312/ “X2 = 2.5276 p=0.111868)” – meaningless to show this many decimals on statistical tests and p-values, should restrict to three. Correct symbology for X2.

15-320/ Formative Period cranial depressions are said to show average diameter of 1.3 - 1.8 cm. Would be more useful to present this as the mean ± SD (or median if more appropriate) and comment on whether these are statistically smaller than the average of ca. 2 cm noted for the Archaic. Ditto LIP.

18-390/ “And of the individuals affected with trauma, 40% (n=4/10) showed perimortem

trauma (S2 Fig), most causing immediate death.”

In a forensic sense, it is not possible to state that even severe unhealed skeletal trauma were the cause of death, as this may have been brought about by another injury that is not visible.

Many researchers would express caution in relating fractures to both the ulna and radius as the result of warding off a blow, as in most cases the ulna alone is affected and dissipates the blow preventing it from affecting the radius. Falls may be implicated instead.

19/ Odd to interpret human 87Sr/86Sr values ranging from 0.7090 to 0.7085 as “very homogeneous marine values”. The mean of the three marine samples provided in Table 2 is 0.70915 ± 0.00001, which is ~consistent with what is expected for the modern ocean. Even the high end of the human range, 0.7090, is slightly but significantly lower than this. These are not pure marine values as is implied but rather show a varying, if generally low, degree of terrestrial Sr input. But the balance between the two depends on the proximity of the terrestrial faunal valley samples to the coastal archaeological sites sampled, i.e., is the near-coastal terrestrial environment around the sites adequately represented? Baseline sample locations are not provided or shown on Figure 1.

Table 2, faunal 87Sr/86Sr values separated by comma rather than period as for the humans (should be consistent). Presumably ‘Actual’ is ‘Modern’. Falange should be Phalanx.

26-480/ “sling stone have not been recorded for the Archaic periods” > used in the singular throughout, shouldn’t it be plural? (and so match the verb).

28-527/ “is interpreted like expression of personal violence” > grammar

29-535/ “most of the affected individuals died almost immediately” > cannot say this based on the lack of healing, which will take some weeks to manifest

29-544/ “To cause these lethal traumas with a mace, opponents had to be facing one another in close physical contact, delivering accurate blows, first targeting the legs, causing the opponent to fall to the ground, before the perpetrator crushing their skull and annihilating them.”

This seems too prescriptive an account. While targeting the legs would be one tactic, a heavy blow with a mace against any part of the body could incapacitate. Were any such blows to the legs recorded? None are mentioned; in fact the methodology refers only to recording trauma to the upper body. So this is completely unsupported. Also unnecessary to refer to ‘annihilating’ the victim.

30-562/ “… we can affirm that these hunting weapons were also used as lethal weapons to slaughter some individuals.”

‘slaughter’ is a rather loaded term; is it warranted/necessary to describe this injury? The circumstances of the encounter are of course unknown – this individual may have been the aggressor who was killed with the implement the intended victim had to hand. Would this then be termed a ‘slaughter’? More factual/neutral language should be considered.

32-607/ “On the other hand, all individuals affected with lethal trauma correspond to young adults”

Needs to be seen in the context of children and adolescents being excluded from the study. Also, as noted above, that most cranial injuries are described as well healed suggests that these younger age groups may have been involved in violence.

32-609/ “However, we also identified one female in the Formative period with a perimortem mandibular fracture, and two females in the LIP who must have died from high impact blows to the head. This shows that lethal violence increased against female individuals in the late periods.”

This conclusion is made on the basis of three individuals? Not warranted.

32-613/ “trauma due to interpersonal violence occurs primarily against individuals between the ages of 20 and 35.”

Given that most examples are healed, they have have occurred many years beforehand. The exclusion of younger individuals from the study must raise questions over this conclusion. There are many examples of violence-related lethal trauma to subadults in the literature.

32-618/ The statement that 95% of the analysed individuals were “concordant with the Sr

chemical signature of the marine fauna, both archaeological and present” should be re-considered. Given the precision of the method, even values of 0.9070 are inconsistent with a purely marine signal (ca. 0.90715), and most of the reported human values in Table 2 are lower than this. Even if these individuals are all ‘local’ to the coast, plant foods with lower 87Sr/86Sr values must be contributing to diets. It is not clear exactly where the terrestrial baseline fauna come from, and so it is difficult to say at what distance from the coast marine values (i.e., from sea-spray) cease to be a factor.

33-633/ “Only from the Late Archaic onward, identify one individual” > grammar

34/ the interpretation of the 2 ‘non-local’ Formative Periods individuals with trauma from Playa Miller 7 is inconsistent. They are first said to have been “integrated into the coastal community” but then the same evidence is used to conclude that “The presence of these two non-coastal individuals suggests the possibility of hostile social interactions between the fishers and their close neighbors”

34/ “Moreover, some individuals were buried naked, without grave goods” (see also Figure 9) – the absence of grave goods does not imply that they were buried naked. Is textile preservation ubiquitous? Were the skeletons recovered from controlled archaeological excavations, or in the aftermath of looting that sadly affects many sites in northern Chile?

36/ reference is made to rock art sites 3000-4000 m asl, whereas the method section states: we analyzed the images displayed in rock art (engravings and paintings) and geoglyphs located along the coast and valleys (up to 2000 m above sea level)”

37/ “10.000 years in which they lived without contact with the West.”

Briefly explain why this is relevant (i.e., with reference to Ferguson and Whitehead’s 1992 ‘War in the tribal zone’, which is presumably what it is being evoked)

39-783/ “the self-consumption of coastal groups” > oddly phrased, suggest ‘subsistence’

41-818/ “no comparable to the large shell middens” > grammar

41-821/ “The reason for inhabiting this open and exposed terrace, which clearly offered its dwellers no natural protection features, were probably…” > grammar

41-825/ “Perhaps the explanation for why was less inhabited than the southern coast had to do with the lack” > grammar

43/ conflictiveness > not recognised as a word by the OED, and awkward regardless

Reviewer #2: This paper investigates long-term trends in skeletal evidence for inter-personal violence among coastal societies of northern Chile. Strontium isotopes are used to determine whether individuals are of local local or non-local groups. The question of diachronic trends in interpersonal violence is of broad interest, and the combination with isotopic analyses represents a novel approach to better contextualize the skeletal evidence for trauma and to allow for more nuanced interpretations of social variation in the past.

My most significant concern with the paper relates to the theoretical framing of violence in human societies. The authors very briefly review the nature of violence on lines 68 to 77 – as being either – “anchored in biology and molded by evolution”, OR ‘explained by levels of social complexity’. They present this as an ‘either/or’ scenario, which is a false dichotomy which neglects the vast majority of literature in bioarchaeology and mischaracterizes the theoretical landscape around violence. The ‘violence is hard wired evolutionarily in humans’ trope is lazy scholarship at best, populist evolutionary psychology story telling at worst. By ‘evolutionary’ do the authors mean that there is long-term change over time? If that is the case, what do they mean by evolutionary? Is all change evolution? Where is the natural selection? What genes might be involved? How does it relate to survivorship or differential reproductions? Many systems change in ways that are non-adaptive and contingent on natural or cultural environments. The current paper really has nothing to say about this debate and adopts and evolutionary framework uncritically. Even if one accepts the ‘violence is hard wired’, then why would anyone be interested in bioarchaeological variation through time? The answer is that there may be a whole range of underlying social, demographic, and environmental factors which lead to different rates of interpersonal violence through time. Disentangling these factors constitutes the most significant body of research in bioarcheology and is what this paper is actually trying to do, yet the authors have failed to cite any of this literature. The fact that interpersonal violence is present in all human societies at relatively minor levels is well documented bioarcheologically (see edited volumes by Martin et al., Knusel et al for example). The interesting question, and the real question posed by this paper, is whether there are different frequencies in populations through time and space, and what factors might explain that variation. Instead, they frame violence as explained either by evolution (a question that cannot be answered by the current study), or that it ‘may be explained by social complexity’. Yes, that is a possibility but also another ‘meta-theory’ but it also completely ignores any other aspects of local/regional variation that may contribute to the expression of violence. To pick two overly simplistic and diametrically opposed theories to characterize the theory around interpersonal violence and ignore all of the subtle and informative research that characterizes most of the literature is intellectually bereft as neither theory actually addresses the complexity of context which underlies variation in expression. More importantly, it fails to adequately frame what this paper actually tests.

I note that the current paper looks at diachronic trends that relate to periods of major social change from fishing economies to chiefdoms. The strength of the work involves the integration of other lines of evidence to test hypotheses about possible correlates of interpersonal violence: local/migrant status of individual, settlement patterns including the presence of ‘defensive’ architecture and location of dwellings, rock art depictions of violence, funerary inclusions, and weaponry. The interpretations in the discussion revolve around interpretation of the relationships between trauma frequencies and these other lines of archaeological evidence. Why is the paper not framed around questions that relate to these lines of evidence? To frame the paper around evolution (which is untestable in the current study) and not pose questions that relate to the relationship between violence and culture (which can and is tested in the current study) both mischaracterizes the research and is a missed opportunity for clarity and impact of the work.

The results that show that there is no significant difference in Sr isotopic values between those with and without trauma is unsurprising since there are so few individuals of non-local origin (only 5 individuals) within the overall sample. When these are then broken down into temporal groups the statistical comparisons are even less informative since some groups contain only one individual. I would be cautious about interpretations of these results.

A particularly interesting observation of the study is that there appears to be a decrease of markers of interpersonal violence perpetrated against males over time, while it remained relatively invariant on females. I think this result warrants further discussion and framing in terms of the articulation of research questions at the beginning of the paper.

Overall, I recommend reorientation of the paper around clear research questions that can be addressed with the data that is used in the paper. This would involve either removal of the untestable ‘evolutionary’ framework, or simple acknowledgement that the debate exists but that the current work cannot directly address those larger evolutionary questions and will focus on whether interpersonal violence corresponds with specific cultural variants over 10ky of prehistory.

Minor corrections

Abstract: How can the absence of something (centralized political power) explain the presence of interpersonal violence? Please clarify or expand.

Lines 562-569 – describes an individual from the formative period where a lithic point is embedded in the neural arch and concludes that the individual would have been paralyzed and that death occurred shortly after impact. I think that to discuss paralysis it implies survival for some time, so unless there is stronger evidence for a period of survivorship I would remove this interpretation.

Line 50 – delete ‘propose to’

Line 476 – I think ‘preferably’ should be changed to ‘most frequently’

Line 408 - stone should be plural

Line 787 – costal should be coastal

6. PLOS authors have the option to publish the peer review history of their article (what does this mean?). If published, this will include your full peer review and any attached files.

Reviewer #1: No

Reviewer #2: No

---

## [Author Response · Author response to Decision Letter 0]

19 May 2023

Arica, February 12, 2023

Reply to editor and reviewers

Reply to editor´s comments

Comment 1: We noted in your submission details that a portion of your manuscript may have been presented or published elsewhere. [Data on violence among Archaic period Chinchorro hunters and fishermen from the far northern coast of Chile were published in American Journal Physical Anthropology, Standen et al. 2020, doi: 10.1002/ajpa/.24009. The novelty of this new Manuscript is that it assesses the long-term trajectory of violence in societies that inhabited the coast of the Atacama Desert in northern Chile from the Archaic Period to the pre-European Contact Period, i.e. for 10,000 years.] Please clarify whether this publication was peer-reviewed and formally published. If this work was previously peer-reviewed and published, in the cover letter please provide the reason that this work does not constitute dual publication and should be included in the current manuscript.

Reply: The article Standen et al. (2020) was published in the AJPA (DOI: 10.1002/ajpa/.24009), following external peer review. The article focuses, exclusively, on expressions of violence in the skeletal record of the Archaic period (10,000–4,000 cal yr BP), Chinchorro culture. For this PLosOne manuscript, we only took its quantitative data (percentages of violence, n = 136 individuals) and the results of strontium analysis (n = 50). This manuscript focuses on a long-term comparative view that includes the Archaic (Chinchorro Culture), Formative and Late Intermediate periods (10,000 BP - AD 1436); totaling 288 individuals. Furthermore, 58 new samples were added to the strontium analysis results database, totaling 108 specimens, including all the indicated cultural periods. Most relevant is that the bioarchaeological data were contextualized with rock art data, artifacts possibly used as weapons, and settlement patterns, all of which gives greater interpretative strength, and a more comprehensive view of the anthropological problem of violence through time.

Comment 2. We note that Figure 1 in your submission contain [map/satellite] images which may be copyrighted. All PLOS content is published under the Creative Commons Attribution License (CC BY 4.0), which means that the manuscript, images, and Supporting Information files will be freely available online, and any third party is permitted to access, download, copy, distribute, and use these materials in any way, even commercially, with proper attribution. For these reasons, we cannot publish previously copyrighted maps or satellite images created using proprietary data, such as Google software (Google Maps, Street View, and Earth). For more information, see our copyright guidelines: http://journals.plos.org/plosone/s/licenses-and-copyright. We require you to either (1) present written permission from the copyright holder to publish these figures specifically under the CC BY 4.0 license, or (2) remove the figures from your submission:

Please upload the completed Content Permission Form or other proof of granted permissions as an "Other" file with your submission.In the figure caption of the copyrighted figure, please include the following text: “Reprinted from [ref] under a CC BY license, with permission from [name of publisher], original copyright [original copyright year].”

Reply: Agree and Fixed. Images of Figure 1a y 1b, were replaced by ink line maps.  

Reply to Reviewers' comments:

Reviewer #1:

Query 1: The context of the study and the questions being posed are well set out, and place in a wider debate concerning the extent of, and variation in, violence in hunter-gatherer societies more generally. In this sense, the paper makes an important contribution. However, there are some issues regarding: (a) the methods employed to address prevalence given the varying state of in/completeness of the skeletons in the sample that needs to be addressed in more detail, even if only in SI. (b) Similarly, there is insufficient information on how the Sr isotope analysis was undertaken.

Reply: Agree and fixed. These issues were resolved in the methodology section. Concerns “a” and “b” are addressed more specifically in queries 10 and 9 respectively.

Query 2: The interpretation of ca. 95% of the human Sr isotope values as being essentially ‘marine’ needs greater nuance, as none are actually consistent with what is expected from a 100% marine signal, but instead show varying terrestrial input: while this may be generally low (and so 'predominantly marine' might be justified) this is not clear from the information provided, which does not include sample locations for the terrestrial faunal baseline (how close to the coast are they?).

Reply: Agreed and fixed. The interpretation of an essentially marine diet is nuanced. However, consider that of 108 individuals sampled (from all chronological periods), 95% (103 individuals) show elevated Sr values, ranging from 0.709176 to 0.708172, which is close to the Sr value for marine fauna; however, the lower values in that range certainly reflect some terrestrial influence. Only 5 individuals show Sr values between 0.706385 and 0.707859, which are closer to the values for terrestrial fauna than for marine fauna. 

Additionally, we incorporate Figure 1b, which shows the localities where wild rodents were captured to establish the strontium baseline in the extreme north of Chile. A paragraph was included in the Material and Methods section indicating the localities sampled and the distance from the coast.

Query 3: While the paper is generally clear and well written, the language used on occasion is overly dramatic and unwarranted based on the evidence alone (e.g., reference to ‘annihilation’ and ‘slaughter’). Some of the images of soft tissue may be seen as problematic for the indigenous (or those claiming descent) communities of the region (e.g., because of their potential sensitivity, Society for American Archaeology journals would not accept these images, or indeed any images of human remains or funerary objects, without express written consent from representatives of the most appropriate source communities. While this may not be the policy of PLoS ONE, the authors and editors should be aware of this potential issue).

Reply: Agreed and fixed. The referred terms “annihilation” and “slaughter” were deleted. The second one was replaced by the term killing. Regarding Figure 3 that shows soft tissue and explicit evidence of trauma, we point out that in the study region there are no native coast peoples that claim a direct link with these prehispanic populations.

Query 4: 4-92/ Confusing to move from (cal) BP to AD in the cultural periodisation. Suggest one or the other.

Reply: We keep the acronyms BP for dates before the Christian era and AD for dates after, since they are accepted and used in other articles published in PlosOne (see for example Nagaoka et al. 2017; DOI:10.371/journal.pone.0185421

Query 5: 6-135/ “evidence of interpersonal violence trauma” > grammar

Reply: Agree and fixed. The sentence stands as follows: “evidence of trauma caused by interpersonal violence.”

Query 6: Why focus on adults only? To what extent was violence also present in subadults? See p 13 where cranial depressed fractures are said to be shallow due to the “advanced process of bone remodeling that most of them show”, implying that they were inflicted earlier in life. All of the traumas for this group show healing, but would this also be the case if older subadults were included? What is the relationship between adult age and their expression?

Reply: Indeed, the reviewer is correct that some cranial traumas with high degree of bone remodeling may have occurred when the individuals were younger (e.g., sub-adults). However, the fragility of the bones of individuals at an early age makes it difficult to record evidence of such characteristics. This is evident in the specialized literature, which concentrates mainly on adult individuals. In addition, methodological limitations of the osteological record do not allow us to know how long before the death of the individual the acute trauma occurred.

Query 7:10-210/ “the presence of fractures in butterfly-shaped was decisive” > grammar

Reply: Agreed and fixed. The sentence stands now us follow: “the fractures in butterfly-shape support the idea to consider them intentional perimortem fractures.”

Query 8: While strontium isotope analysis may fall within the broad remit of ‘geoarchaeology’, the fit is perhaps not the best. Suggest 'biogeochemistry' (not a required revision, a suggestion only).

Reply: We maintain the term geoarchaeology, because is commonly use in the discipline.

Query 9: The Methods section on Sr is inadequate. There is no information on sample pretreatment or analysis, including whether measurements were made using TIMS or MC-ICP-MS, which impacts on their precision.

Reply: Agreed and fixed. Additional information on this issue was added in the Material and Methods section, subsection Geoarchaeology: Strontium (Sr) isotopes. The whole paragraph stands as follows: 

Processing the samples: All samples for Sr isotope geochemistry were processed in the radiogenic isotope geochemistry laboratory at the University of North Carolina - Chapel Hill. Superficial contamination on samples was removed by ultrasound cleaning and light polishing with a motorized polishing bit. The same tool with a diamond bit was used to powder 2-3 mg of tooth enamel (human samples) or bone (animals). All samples were dissolved in 3.5 M HNO3 and Sr was isolated from the samples using standard column chromatographic techniques. One microliter of concentrated H3PO4 was added to the Sr solution and the samples were evaporated to a small drop for analysis. All samples were loaded onto single Re filaments with TaCl5 and loaded into the VG-Sector-54 thermal ionization mass spectrometer at Chapel Hill. Analyses were done using triple-dynamic multicollector mode with 88Sr—3 V (10−11 Ω resistor). Data data were normalized to 86Sr/88Sr = 0.1194 assuming exponential mass fractionation behavior. Data are reported relative to NBS-987 87Sr/86Sr = 0.710250. Replicate analyses of the standard over the period of this study yielded and uncertainty in the 87Sr/86Sr of ± 0.000010 (n = 12).

Query 10: 13-272/ “(independently of the degree of body completeness)”. Explain further how complete vs. incomplete skeletons were analysed, as this is crucial to providing population estimates of trauma prevalence. This is evident from the different prevalence figures provided for only those individuals ‘recovered with the cranium’ [complete?] which present the majority of traumas relateable to interpersonal violence. Ditto for the later periods. Usually it is stated that a certain proportion of the element in question had to be present for it to ‘count’. Should also address how the 20% of individuals (which is more than a ‘few’, p 6) with soft tissues were treated, given that – depending on the degree of soft tissue preservation – the skeleton would not be as visible for macroscopic observation, e.g., of embedded projectile points/fragments. 

Reply: Agreed and fixed. Explanation for this issue was included in the Material and Method section, subsection “The Bioarchaeological Sample”, and the text stands as follows: “The total sample of individuals was separated in two categories according to their degree of completeness: (a) individuals with only the cranium preserved, and (b) individuals completed or only post-craniun preserved. The first subcategory corresponds only to skulls with trauma; the second category, counted independently of the first, and corresponds to postcranial trauma.” 

Additionally, individuals with soft tissue remains (ca. 20% of the total sample studied), allowed us to visualize traumas that only affected these tissues and were added to the total number of individuals with trauma analyzed. Some of the cases with more complete soft tissue were X-rayed, which did not show bone trauma.

Query 11: 13-303/ a minor point, but the age of individual MEC/C1 is given as 16-17 years, whereas the Methods section states that adults in the study are defined as >17 years.

Reply: Agree and fixed. The age was corrected to 17-18 years old.

Query 12: 312/ “X2 = 2.5276 p=0.111868)” – meaningless to show this many decimals on statistical tests and p-values, should restrict to three. Correct symbology for X2.

Reply: Agree and fixed. The p-values and symbology stand now as follows X2 = 2.527 p=0.112, and it was corrected in the whole text.

Query 13: 15-320/ Formative Period cranial depressions are said to show average diameter of 1.3 - 1.8 cm. Would be more useful to present this as the mean ± SD (or median if more appropriate) and comment on whether these are statistically smaller than the average of ca. 2 cm noted for the Archaic. Ditto LIP.

Reply: Agreed and fixed. Due to the reduced number of cases, the maximum and minimum values of the dimensions of the fractures of the cranial vault depression were presented.

Query 14: 18-390/ “And of the individuals affected with trauma, 40% (n=4/10) showed perimortem trauma (S2 Fig), most causing immediate death.”? In a forensic sense, it is not possible to state that even severe unhealed skeletal trauma were the cause of death, as this may have been brought about by another injury that is not visible.

Reply: Agreed and fixed. The sentence -most causing immediate death- was delete.

Query 15: Many researchers would express caution in relating fractures to both the ulna and radius as the result of warding off a blow, as in most cases the ulna alone is affected and dissipates the blow preventing it from affecting the radius. Falls may be implicated instead.

Reply: Agree ad fixed. The whole sentence was rewritten and stands as follows: “When this is the case, the attacked individual instinctively reacts by protecting his face with his hands and forearms, so that the impact of the blows is received directly on the ulna. Although we consider that these traumas reaffirm the idea of body-to-body confrontations, these injuries could have been also caused by accidental falls or in contexts of fights”.

Query 16: 19/ Odd to interpret human 87Sr/86Sr values ranging from 0.7090 to 0.7085 as “very homogeneous marine values”. The mean of the three marine samples provided in Table 2 is 0.70915 ± 0.00001, which is ~consistent with what is expected for the modern ocean. Even the high end of the human range, 0.7090, is slightly but significantly lower than this. These are not pure marine values as is implied but rather show a varying, if generally low, degree of terrestrial Sr input. But the balance between the two depends on the proximity of the terrestrial faunal valley samples to the coastal archaeological sites sampled, i.e., is the near-coastal terrestrial environment around the sites adequately represented? Baseline sample locations are not provided or shown on Figure 1.

Reply: Agree ad fixed..

In Fig. 1a, a map showing the location of the sites sampled for our strontium baseline was added, including marine and terrestrial fauna. Besides, in the methology section the following text was added for clarification:

“Sampling localities: The Sr isotopic baseline was elaborated from the animal model for the extreme north of Chile (~18°30'-19°40' S). Bone samples of modern and archaeological fauna, both marine and terrestrial, were taken from 10 localities located from sea level to 4,200 masl (Figure 1b). Modern wild rodent fauna (n=12) were captured with Sherman traps that were placed in uninhabited areas to ensure that rodents had exclusively consumed local grasses. They were captured in: (a) the mouths of the Arica and Camarones rivers, exactly 2 km from the current beach line and at 0 masl; (b) the desert oasis (Tiliviche and Conanoxa) 30-40 km from the coast respectively and at 900 masl; the foothills (Patapatane and Putre) 70 km from the coast and at 3,700 masl; and the altiplano (Hakenasa, Las Cuevas, Lauca, Colchane) between 4,000 - 4,400 masl and 130 - 150 km from the coast (Figure 1b). In addition, camelid bone samples (n=6) were collected at six Archaic archaeological sites, distributed in the same area where the present-day fauna samples were taken (Figure 1b; Table 2). The marine mammal bone samples (n=3), correspond to a sea lion found dead on the shore of the beach (Arica coast) and two archaeological samples (whale rib from the Arica coast; and sea lion from Tiliviche (which must have been moved from the coast to the interior oasis by Archaic populations)”.

Regarding the interpretation of the marine signal from pre-Hispanic coastal populations, it should be noted that wild rodents captured at the mouth of the rivers of the Arica (Lluta river) and Camarones coast in the same environments that inhabited the coastal populations and, therefore, possibly affected by the effect of seawater splashing, gave significantly lower Sr values (87Sr/86Sr 0.707402 - 0.707210) than coastal humans (87Sr/86Sr 0.709176 - 0.708583), but more similar to values for wild rodents from desert oases (87Sr/86Sr 0.707266 - 0.706536; 30-40 km inland from the coast).

Query 17: Table 2, faunal 87Sr/86Sr values separated by comma rather than period as for the humans (should be consistent). Presumably ‘Actual’ is ‘Modern’. Falange should be Phalanx.

Reply: Agree and fixed. Many thanks for noticing these fine details. “Actual” was replaced by “modern”.

Query 18: 26-480/ “sling stone have not been recorded for the Archaic periods” > used in the singular throughout, shouldn’t it be plural? (and so match the verb).

Reply: Agree and fixed. The plural and singular forms of the terms period and sling stone were reviewed.

Query 19: 28-527/ “is interpreted like expression of personal violence” > grammar.

Reply: Agreed and fixed. The sentence stands now as follows: “is interpreted as an expression of interpersonal violence”.

Query 20: 29-535/ “most of the affected individuals died almost immediately” > cannot say this based on the lack of healing, which will take some weeks to manifest.

Reply: Agree and fixed. The sentence stands now as follow: “In relation to the perimortem traumas, the target was the cranium and thoracic cavity, which could have caused the death of the individuals.”

Query 21: 29-544/ “To cause these lethal traumas with a mace, opponents had to be facing one another in close physical contact, delivering accurate blows, first targeting the legs, causing the opponent to fall to the ground, before the perpetrator crushing their skull and annihilating them.”? This seems too prescriptive an account. While targeting the legs would be one tactic, a heavy blow with a mace against any part of the body could incapacitate. Were any such blows to the legs recorded? None are mentioned; in fact the methodology refers only to recording trauma to the upper body. So this is completely unsupported. Also unnecessary to refer to ‘annihilating’ the victim.

Reply: Agree and fixed. These sentences were deleted.

Query 22: 30-562/ “… we can affirm that these hunting weapons were also used as lethal weapons to slaughter some individuals.” ? ‘slaughter’ is a rather loaded term; is it warranted/necessary to describe this injury? The circumstances of the encounter are of course unknown – this individual may have been the aggressor who was killed with the implement the intended victim had to hand. Would this then be termed a ‘slaughter’? More factual/neutral language should be considered.

Reply: Agree and fixed. The phrase stands as follows: “we suggest that these hunting weapons could also have been used as lethal weapons.”

Query 23: 32-607/ “On the other hand, all individuals affected with lethal trauma correspond to young adults” ? Needs to be seen in the context of children and adolescents being excluded from the study. Also, as noted above, that most cranial injuries are described as well healed suggests that these younger age groups may have been involved in violence.

Reply: Agree and fixed. The statement that only young adults were affected with lethal trauma was rephrased in the text. This is considering that subadult individuals were not studied, so we cannot know whether or not they were affected with lethal trauma (see also reply to Query 25).

Query 24: 32-609/ “However, we also identified one female in the Formative period with a perimortem mandibular fracture, and two females in the LIP who must have died from high impact blows to the head. This shows that lethal violence increased against female individuals in the late periods.”? This conclusion is made on the basis of three individuals? Not warranted.

Reply: Agree and fixed. The sentence stands now as follows: “These data, although scarce, raise the question of whether lethal violence against women increased in the late periods.”

Query 25: 32-613/ “trauma due to interpersonal violence occurs primarily against individuals between the ages of 20 and 35.? Given that most examples are healed, they have have occurred many years beforehand. The exclusion of younger individuals from the study must raise questions over this conclusion. There are many examples of violence-related lethal trauma to subadults in the literature.

Reply: The reviewer is correct in stating that the young individuals in the sample were the most affected by the trauma, and given that most of them are healed, the trauma must have occurred when the individuals were even younger. However, since bone tissue regeneration times vary depending on the location of the trauma, the bone affected, the type of trauma, among other variables, it is not possible to determine precisely at what age the individual was exposed to acute trauma. Therefore, we did not delve into this aspect of violence. In future research we will focus on the study of subadult collections (infants, children and adolescents) to address this issue.

Query 26: 32-618/ The statement that 95% of the analysed individuals were “concordant with the Sr chemical signature of the marine fauna, both archaeological and present” should be re-considered. Given the precision of the method, even values of 0.9070 are inconsistent with a purely marine signal (ca. 0.90715), and most of the reported human values in Table 2 are lower than this. Even if these individuals are all ‘local’ to the coast, plant foods with lower 87Sr/86Sr values must be contributing to diets. It is not clear exactly where the terrestrial baseline fauna come from, and so it is difficult to say at what distance from the coast marine values (i.e., from sea-spray) cease to be a factor.

Reply: The reviewer is correct in pointing out that the diets of coastal populations included the consumption of plants from the valleys and quebradas. This may include wild plants during the Archaic Period (e.g., fruits of opuntia, prosopis, scirpus), and wild and cultivated plants during the Formative and Late Intermediate Periods (e.g., squash, maize, chilies, among others). However, the delta of strontium values of coastal humans is closer to the reference pattern of marine fauna than that of coastal, desert oasis and altiplano terrestrial fauna. This contrast can also be seen in Figure 5, so we maintain our more nuanced interpretation of the importance of the marine diet in coastal populations. This trend of marine diets among coastal populations has been emphasized with Nitrogen and Carbon stable isotope studies (Alfonso et al. 2019; King et al. 2018; Robert et al. 2013). To broaden and deepen this point, Figure 1b was incorporated, where the sampled localities from northernmost Chile are indicated, on which the strontium stable isotope reference pattern was elaborated, based on the animal model (see response to Query 16).

Query 27: 33-633/ “Only from the Late Archaic onward, identify one individual” > grammar

Reply: Agree and fixed. The phrase stands as follows: “In the Late Archaic, one individual (M1T27C18, male) was identified with a non-marine Sr chemical signature, showing that he came from another geological locality.”

Query 28: 34/ the interpretation of the 2 ‘non-local’ Formative Periods individuals with trauma from Playa Miller 7 is inconsistent. They are first said to have been “integrated into the coastal community” but then the same evidence is used to conclude that “The presence of these two non-coastal individuals suggests the possibility of hostile social interactions between the fishers and their close neighbors”

Reply: Agree and fixed. The phrase “suggesting they were also integrated into the coastal community” was deleted to make the text more consistent.

Query 29: 34/ “Moreover, some individuals were buried naked, without grave goods” (see also Figure 9) – the absence of grave goods does not imply that they were buried naked. Is textile preservation ubiquitous? Were the skeletons recovered from controlled archaeological excavations, or in the aftermath of looting that sadly affects many sites in northern Chile?

Reply: The good preservation of organic remains in the Atacama Desert allows the textiles or part of them to be very well preserved (see, for example, Figure 8). Thus, the individuals illustrated in Figure 9 were buried without any type of clothing. As for the provenance of the studied collections, some of them were generated by controlled excavations of salvage during the 1970s and 1980s. However, at that time archaeologists did not publish mortuary contexts that were not aesthetically museum-worthy. This bias in excavations may have made past instances of violence even more invisible.

Query 30: 36/ reference is made to rock art sites 3000-4000 m asl, whereas the method section states: we analyzed the images displayed in rock art (engravings and paintings) and geoglyphs located along the coast and valleys (up to 2000 m above sea level)”

Reply: Indeed, in the methods section, only sites with rock art near the coast were considered, where Formative and PIT populations may have participated in its execution. In contrast, rock art from the Archaic period has been identified mainly in the Sierra and the Altiplano, showing camelid hunting scenes, and its executors would have been Andean hunters and gatherers.

Query 31: 37/ “10.000 years in which they lived without contact with the West.”? Briefly explain why this is relevant (i.e., with reference to Ferguson and Whitehead’s 1992 ‘War in the tribal zone’, which is presumably what it is being evoked)

Reply: In the background section we incorporate the following paragraph based on the model of warfare and tribal zone proposed by Ferguson and Whitehouse (1992). The paragraph is as follows: " From the perspective of historical anthropology, Ferguson and Whitehead (1992) proposed the thesis of "tribal zone warfare" for territories or zones of conquest and subjugation by state or imperial societies over politically decentralized native societies. Contact between dominators and subjugated in the tribal zone would have caused a substantial intensification of warfare and violence, due to the introduction of new steel weapons of greater lethality, altering the cultural patterns of native societies. Ferguson and Whitehead (1992) applied this model to understand and interpret the historically and ethnographically documented warfare in the Amazonian lowlands of South America. Our study is also situated in the South American lowlands, specifically in the narrow coastal strip along the Atacama Desert, one of the driest on the planet. The populations studied correspond small-scale, politically decentralized societies, with a scale of 10,000 years of trajectory and prior to contact with the West.”

Query 32: 39-783/ “the self-consumption of coastal groups” > oddly phrased, suggest ‘subsistence’

Reply: Agree and fixed. The word stands now: “subsistence”.

Query 33: 41-818/ “no comparable to the large shell middens” > grammar

Reply: Agree and fixed. The sentence stands now: Another difference is that domestic sites on the north coast correspond to small camps, which contrast with the large shell middens of the south coast that can reach heights of four to five meters of accumulated debris derived from daily life.

Query 34: 41-821/ “The reason for inhabiting this open and exposed terrace, which clearly offered its dwellers no natural protection features, were probably…” > grammar

Reply: Agree and fixed. The sentence stands now: The reason for inhabiting this open and exposed terrace, which evidently offered its inhabitants no natural protection, was due to the diversity of resources that were concentrated at the mouth of this river: drinking water, springs with aquatic plants such as reeds, and a diversity of avifauna, fish and bivalves that inhabited the sandy beaches, very attractive to hunter-gatherers and fishermen.

Query 35: 41-825/ “Perhaps the explanation for why was less inhabited than the southern coast had to do with the lack” > grammar

Reply: Agree and fixed. The sentence stands now: “Perhaps the explanation as to why it was less inhabited than the southern coast may have been due to the lack of natural protection offered by this coast, which, although rich in resources, was unprotected against possible ambushes or surprise attacks”

Query 36: 43/ conflictiveness > not recognised as a word by the OED, and awkward regardless

Reply: Agree ad fixed. The sentence stands now as follows: “but also, of tension and conflict.”

Reviewer #2:

This paper investigates long-term trends in skeletal evidence for inter-personal violence among coastal societies of northern Chile. Strontium isotopes are used to determine whether individuals are of local local or non-local groups. The question of diachronic trends in interpersonal violence is of broad interest, and the combination with isotopic analyses represents a novel approach to better contextualize the skeletal evidence for trauma and to allow for more nuanced interpretations of social variation in the past.

Query 37: My most significant concern with the paper relates to the theoretical framing of violence in human societies. The authors very briefly review the nature of violence on lines 68 to 77 – as being either – “anchored in biology and molded by evolution”, OR ‘explained by levels of social complexity’. They present this as an ‘either/or’ scenario, which is a false dichotomy which neglects the vast majority of literature in bioarchaeology and mischaracterizes the theoretical landscape around violence. The ‘violence is hard wired evolutionarily in humans’ trope is lazy scholarship at best, populist evolutionary psychology story telling at worst. By ‘evolutionary’ do the authors mean that there is long-term change over time? If that is the case, what do they mean by evolutionary? Is all change evolution? Where is the natural selection? What genes might be involved? How does it relate to survivorship or differential reproductions? Many systems change in ways that are non-adaptive and contingent on natural or cultural environments. The current paper really has nothing to say about this debate and adopts and evolutionary framework uncritically. Even if one accepts the ‘violence is hard wired’, then why would anyone be interested in bioarchaeological variation through time? The answer is that there may be a whole range of underlying social, demographic, and environmental factors which lead to different rates of interpersonal violence through time. Disentangling these factors constitutes the most significant body of research in bioarcheology and is what this paper is actually trying to do, yet the authors have failed to cite any of this literature. The fact that interpersonal violence is present in all human societies at relatively minor levels is well documented bioarcheologically (see edited volumes by Martin et al., Knusel et al for example). The interesting question, and the real question posed by this paper, is whether there are different frequencies in populations through time and space, and what factors might explain that variation. Instead, they frame violence as explained either by evolution (a question that cannot be answered by the current study), or that it ‘may be explained by social complexity’. Yes, that is a possibility but also another ‘meta-theory’ but it also completely ignores any other aspects of local/regional variation that may contribute to the expression of violence. To pick two overly simplistic and diametrically opposed theories to characterize the theory around interpersonal violence and ignore all of the subtle and informative research that characterizes most of the literature is intellectually bereft as neither theory actually addresses the complexity of context which underlies variation in expression. More importantly, it fails to adequately frame what this paper actually tests.

Reply: Agree and fixed. This section was entire rewritten and stands as follows: “Several theoretical perspectives based on materialistic, ecological, evolutive and bioarcheological background compete to explain violence among hunter-gatherer societies (Allen and Jones, 2014). The first one estimates that violence and war were not constitutive of the social structure among hunters and gatherers, due to the lack of social and environmental condition to practice it, or to maintain antagonist groups in recurrent confrontation (Ferguson, 1990; Hass, 2001; Fry, 2007). Sahlins (1983) and Cohen (1989), add that lethal violence between individuals of small populations with high mobile residence, could have jeopardized their survival. Conversely, collective violence and war would have been a socio-cultural feature proper of farming societies, concerned with the defense of capital investment (irrigation canal, dams, different form of land preparation, silviculture). Besides defense and protection, in some cases, there was the need for territorial expansion, which was eventually solved by conflicting and competing with foreign social groups (Haas, 2001; Carneiro, 1994). In sum, violence would have increase through time linked to social complexity and population increase. Violence in the ecological perspective stems from the availability of subsistence resources, becoming an adaptive response to deal with their environmental fluctuation (Keeley, 1996; Kelly, 2000; Walker, 2001; LeBlanc and Register, 2003; Otterbein, 2004). Environmental stress coupled with scarcity of subsistence resources should trigger competition and hostile behavior among human groups (LeBlanc and Register, 2003). In contrast, buoyance periods, with greater availability of resources, conflicts and violence would diminish, and banquet and other social expression (funeral and festive ceremonies) could have taken place (LeBlanc and Register, 2003). For the evolutionary perspective, violence and war would be anchored in the biology of our species and shaped through its evolution, which would mean that this behavior would have accompanied Homo sapiens in its long and complex evolutionary trajectory [3-10] (Leroi-Gourhan, 1985; Wrangham and Peterson, 1996). Thus, the capacity for aggressive behavior would be inherent to the human condition, and its evolution shaped by natural selection (e.g., van der Dennen and Falger, 1990; Bowles, 2009; Daly, 2015). In this theoretical view, however, specific patterns of violence would be, also, generated and shaped by external, environmental and sociocultural factors, which are the most evident. These factors would have a greater or lesser influence on the existence, intensity and effects of violence in people's lives. For the bioarchaeological perspective and social theory, violence and war would be codified by intricate cultural meanings (Martin et al., 2012; Nielsen and Walker 2009; Pérez, 2012), which implies that social and symbolic factors would channel different forms of violence, from their own cultural logics, which can be identified and interpreted from the skeletal remains. For prehistoric populations that governed their lives without written records, certain behaviors that from a Western perspective can be seen as violent, irrational, aberrant or deviant, could have had totally different cultural meanings. In other words, violence should not be considered a simple reaction or response to external factors (e.g., environmental stress, competition for resources, population growth), nor would it derive from an innate human biological condition. Social and cultural contexts give violence its power and meaning, which means that different forms of violence would be tolerated, and its use would be culturally legitimized for social control and stability. These approaches have argued that functionalist models of violence, such as those discussed in the preceding paragraphs, limit interpretations of violence in its social and symbolic dimensions (Pérez, 2012; Milner, 2016). [see 3, 11, 12] From the perspective of historical anthropology, Ferguson and Whitehead (1992) proposed the thesis of "tribal zone warfare" for territories or zones of conquest and subjugation by state or imperial societies over politically decentralized native societies. Contact between dominators and subjugated in the tribal zone would have caused a substantial intensification of warfare and violence, due to the introduction of new steel weapons of greater lethality, altering the cultural patterns of native societies. Ferguson and Whitehead (1992) applied this model to understand and interpret the intense warfare ethnographically documented in the Amazonian lowlands of South America. Although our study is also situated in the South American lowlands, specifically in the narrow coastal strip along the Atacama Desert, one of the driest on the planet; the populations studied correspond to politically decentralized societies, with a scale of 10,000 years of development prior to interaction with state or imperial societies. Ferguson and Whitehead's ethnographic and historical approach contrasts with our bioarchaeological approach, which demonstrates that expressions of violence did not occur under the conditions proposed for Amazonia.”

Query 38: I note that the current paper looks at diachronic trends that relate to periods of major social change from fishing economies to chiefdoms. The strength of the work involves the integration of other lines of evidence to test hypotheses about possible correlates of interpersonal violence: local/migrant status of individual, settlement patterns including the presence of ‘defensive’ architecture and location of dwellings, rock art depictions of violence, funerary inclusions, and weaponry. The interpretations in the discussion revolve around interpretation of the relationships between trauma frequencies and these other lines of archaeological evidence. Why is the paper not framed around questions that relate to these lines of evidence? To frame the paper around evolution (which is untestable in the current study) and not pose questions that relate to the relationship between violence and culture (which can and is tested in the current study) both mischaracterizes the research and is a missed opportunity for clarity and impact of the work.

Reply: Texts that gave the impression that the manuscript followed an evolutionary theoretical framework to explain the emergence and evolution of violence were eliminated or paraphrased. It is necessary to point out that the manuscript, from its origin, did not posit that the trajectories of violence in the history of the Atacama Desert depended on genetic factors. On the contrary, our research problem and the considered explanatory factors are inscribed in sociocultural and environmental aspects (rock art, weapons in mortuary contexts and settlement pattern).

A whole paragraph was added at the end of the Introduction that stands as follows: “These possible trajectories were contrasted with other socio-cultural manifestations, such as rock art, weapons in mortuary contexts and settlement patterns, as expressions of violent behavior. In other words, violence as a social phenomenon can be observed not only through the traces left by confrontations on the human body, but also through other expressions of material culture such as those mentioned above.”

Query 39: The results that show that there is no significant difference in Sr isotopic values between those with and without trauma is unsurprising since there are so few individuals of non-local origin (only 5 individuals) within the overall sample. When these are then broken down into temporal groups the statistical comparisons are even less informative since some groups contain only one individual. I would be cautious about interpretations of these results. 

Reply: Agreed and fixed. In different parts of the manuscript this point was carefully refined.

Query 40: A particularly interesting observation of the study is that there appears to be a decrease of markers of interpersonal violence perpetrated against males over time, while it remained relatively invariant on females. I think this result warrants further discussion and framing in terms of the articulation of research questions at the beginning of the paper.

Reply: Agree and fixed. A whole paragraph was added at the end of the Introduction that stands as follows: In addition, we assessed whether interpersonal violence affected males and females equally over time or, conversely, whether violence affected one sex more than the other.

Query 41: Overall, I recommend reorientation of the paper around clear research questions that can be addressed with the data that is used in the paper. This would involve either removal of the untestable ‘evolutionary’ framework, or simple acknowledgement that the debate exists but that the current work cannot directly address those larger evolutionary questions and will focus on whether interpersonal violence corresponds with specific cultural variants over 10ky of prehistory.

Reply: The evolutionary framework was nuanced, presenting it as one of the options for understanding and explaining violent behaviors in the past, but it does not constitute the theoretical underpinning of this study. On the contrary, as we pointed out in the answer to query 38, our study emphasizes that behaviors and expressions of violence are always mediated by sociocultural patterns.

Minor corrections

Query 42: Abstract: How can the absence of something (centralized political power) explain the presence of interpersonal violence? Please clarify or expand.

Reply: Agree and fixed. The corresponding section in the summary has been reworded.

Query 43: Lines 562-569 – describes an individual from the formative period where a lithic point is embedded in the neural arch and concludes that the individual would have been paralyzed and that death occurred shortly after impact. I think that to discuss paralysis it implies survival for some time, so unless there is stronger evidence for a period of survivorship I would remove this interpretation. 

Reply: Agree and fixed. This controversial sentence has been deleted: “This must have resulted in paralysis of the individual’s lower extremities, making any movement impossible.”

Query 44: Line 50 – delete ‘propose to’ 

Reply: Agree and fixed.

Query 45: Line 476 – I think ‘preferably’ should be changed to ‘most frequently’ 

Reply: Agree and fixed.

Query 46: Line 408 - stone should be plural 

Reply: Agree and fixed.

Query 47: Line 787 – costal should be coastal 

Reply: Agree and fixed.

---

## [Decision Letter · Decision Letter 1]

26 Jun 2023

PONE-D-22-28217R1Violence in fishing, hunting, and gathering societies of the Atacama Desert coast: A long-term perspective (10,000 BP - AD 1436).PLOS ONE

Dear Dr. Standen,

Thank you for resubmitting your manuscript to PLOS ONE. Reviewer 1 recommends to accept the manuscript. However, reviewer 2 as indicated below, still has some concerns. The main one has to do with the discussion section which indeed requires editing and a better focus, The length of the discussion section should be cut down to no more than half of its current length or preferably shorter. The aim is however, as indictaed by Revuewer 2, to remove sections which do not discuss aspects which are the direct focus of the paper and deviate from the specified focus in the introduction.

Please resubmit your revised manuscript by Aug 10 2023 11:59PM. If you will need more time than this to complete your revisions, please reply to this message or contact the journal office at plosone@plos.org. Please include the following items when submitting your revised manuscript:A rebuttal letter that responds to each point raised by the academic editor and reviewer(s). You should upload this letter as a separate file labeled 'Response to Reviewers'.A marked-up copy of your manuscript that highlights changes made to the original version. You should upload this as a separate file labeled 'Revised Manuscript with Track Changes'.An unmarked version of your revised paper without tracked changes. You should upload this as a separate file labeled 'Manuscript'.I look forward to receiving your revised manuscript.

Kind regards,

Ron Pinhasi

Academic Editor

PLOS ONE

Journal Requirements:

Reviewers' comments:

Reviewer's Responses to Questions

**Comments to the Author**

1. If the authors have adequately addressed your comments raised in a previous round of review and you feel that this manuscript is now acceptable for publication, you may indicate that here to bypass the “Comments to the Author” section, enter your conflict of interest statement in the “Confidential to Editor” section, and submit your "Accept" recommendation.

Reviewer #1: All comments have been addressed

Reviewer #2: (No Response)

2. Is the manuscript technically sound, and do the data support the conclusions?

Reviewer #1: Yes

Reviewer #2: Partly

3. Has the statistical analysis been performed appropriately and rigorously? 

Reviewer #1: Yes

Reviewer #2: N/A

4. Have the authors made all data underlying the findings in their manuscript fully available?

Reviewer #1: Yes

Reviewer #2: Yes

5. Is the manuscript presented in an intelligible fashion and written in standard English?

Reviewer #1: Yes

Reviewer #2: Yes

6. Review Comments to the Author

Reviewer #1: The authors have addressed the reviewers' comments to my satisfaction. This is a very interesting and useful paper that will provide the basis for further discussion on conflict in the region and more widely.

Reviewer #2: REVISION NOTES:

This paper is significantly improved from my first reading of the manuscript. While I find the revisions mostly effective and I appreciate that the authors have largely removed the false – evolution/culture dichotomy, I still found the wording in places to reflect fairly simplistic views and ‘adaptive story telling’. For example, line 87: “Besides defense and protection, in some cases, there was the need for territorial expansion, which was eventually solved by conflicting and competing with foreign social groups”. By characterizing this as a ‘need’, or as a consequence of ‘capital investment’ the wording implies that power is irrelevant to territorial expansion, but violence a simple and inevitable consequence of ecology and demography rather that social structure and power relations, themes which are well explored in the literature. This point is made later on line 112, which somewhat contradicts this earlier paragraph.

In general I found the discussion to be too long with too many tangents. The authors need to focus the discussion on their research questions. The questions posed by the paper relate to long term trends and whether or not they correspond with isotopic evidence for local or non-local status, yet the discussion is filled with consideration of what types of weapons might relate to different types of trauma, detailed descriptions of how combat may influence trauma patterning, or detailed descriptions of specific individuals, or very detailed discussions of trade and other aspects of cultural context. These sections really have either tenuous links or nothing whatsoever to do with the core questions of the paper, and if anything, detract from the utility of the work. There is no shortage of literature on many tangential topics so it would be sufficient to simply point the reader towards some of this work, or preferably, cut it out altogether. There are 18 pages of discussion in this paper, I think this could be focussed down to a bout 5-6 pages of direct relevance to the research questions. Some of the specific descriptions of burials that are unique to this study could be put into a supplementary file.

Line 85 – delete ‘proper’

Line – 99 Homo sapiens should be italicized

Line 101-103 – awkward wording and too many commas.

Line 133 – needs additional comma after approach, and names of authors of study.

Line 178 – delete ‘would have’, and line 180, should be ‘did violence increase…’

Line 182 – should be in relation to (remove would again).

Line 186 – remove would again… you’re not testing whether anything ‘would’ happen, it’s whether there is evidence that it did happen.

Line 282 – can you provide a citation for the assertion that butterfly shaped fractures in long-bones can be considered to be perimortem, as well as further clarification justifying this assertion.

Line 297 – change wording again from ‘would’, you’re not predicting anything, this is a post-hoc interpretation so a better way to word this is that if you were to see such evidence it ‘could be interpreted as indicative of inter-group violence’

Line 349 - Likewise, our investigate (should be investigation)

Line 577 – I don’t find the argument that unmodified stones in burials may have been weapons to be compelling. Unmodified stones are part of grave fill in just about any burial context anywhere, so this seems to be overly speculative.

Line 681 – change devises to devices

7. PLOS authors have the option to publish the peer review history of their article (what does this mean?). If published, this will include your full peer review and any attached files.

Reviewer #1: No

Reviewer #2: No

---

## [Author Response · Author response to Decision Letter 1]

29 Jul 2023

Dr. Ron Pinhasi

Academic Editor

PLOS ONE

Dear Editor,

Thank you for continuing reviewing our manuscript PONE-D-22-28217R, entitled “Violence in fishing, hunting, and gathering societies of the Atacama Desert coast: A long-term perspective (10,000 BP - AD 1436)”. The rebuttal letter responding to each point raised by the academic editor and reviewer(s) is displayed below.

Response to comments from the editor of Plos One

Query 1: Reviewer 2 as indicated below, still has some concerns. The main one has to do with the discussion section which indeed requires editing and a better focus. The length of the discussion section should be cut down to no more than half of its current length or preferably shorter. The aim is however, as indicated by Reviewer 2, to remove sections which do not discuss aspects which are the direct focus of the paper and deviate from the specified focus in the introduction.

Reply: Agreed and fixed. The discussion section was reduced by 50%. This means that 7 pages were moved to the subsection "trauma, weapons and mode of fighting brawls" in the Results section. We made this decision considering that the data presented in this subsection are significant for reconstructing the modes of interpersonal fighting in the past. At the same time, the remaining text was edited down to 9 pages.

Response to Reviewers'#2 comments:

Query 1: This paper is significantly improved from my first reading of the manuscript. While I find the revisions mostly effective and I appreciate that the authors have largely removed the false – evolution/culture dichotomy, I still found the wording in places to reflect fairly simplistic views and ‘adaptive story telling’. For example, line 87: “Besides defense and protection, in some cases, there was the need for territorial expansion, which was eventually solved by conflicting and competing with foreign social groups”. By characterizing this as a ‘need’, or as a consequence of ‘capital investment’ the wording implies that power is irrelevant to territorial expansion, but violence a simple and inevitable consequence of ecology and demography rather that social structure and power relations, themes which are well explored in the literature. This point is made later on line 112, which somewhat contradicts this earlier paragraph.

Reply: We do not fully agree with the review criticism. We maintained, in part, our approach.

Query 2: In general I found the discussion to be too long with too many tangents. The authors need to focus the discussion on their research questions. The questions posed by the paper relate to long term trends and whether or not they correspond with isotopic evidence for local or non-local status, yet the discussion is filled with consideration of what types of weapons might relate to different types of trauma, detailed descriptions of how combat may influence trauma patterning, or detailed descriptions of specific individuals, or very detailed discussions of trade and other aspects of cultural context. These sections really have either tenuous links or nothing whatsoever to do with the core questions of the paper, and if anything, detract from the utility of the work. There is no shortage of literature on many tangential topics so it would be sufficient to simply point the reader towards some of this work, or preferably, cut it out altogether. There are 18 pages of discussion in this paper, I think this could be focussed down to a bout 5-6 pages of direct relevance to the research questions. Some of the specific descriptions of burials that are unique to this study could be put into a supplementary file.

Reply: Agreed and fixed, see reply to Query 1 regarding the reduction of the number of pages. We also focused the discussion on the main question of the paper: Long-term trends in violence and their correspondence or not with local or extra-local isotopic signatures. We did not move the description of burials to a supplementary file, because we consider that this information is important for the main text.

Query 3, Line 85 – delete ‘proper’.

Reply: Agreed and fixed.

Query 4, Line – 99 Homo sapiens should be italicized

Reply: Agreed and fixed.

Query 5, Line 101-104 – awkward wording and too many commas.

Reply: Agreed and fixed. The sentence stands as follows: Thus, the capacity for aggressive behavior would be inherent to the human condition, and its evolution would be shaped by natural selection [e.g., 15, 20, 21]. In this theoretical view, specific patterns of violence would be generated and shaped by environmental and sociocultural factors

Query 6, Line 133 – needs additional comma after approach, and names of authors of study.

Reply: Agreed and fixed.

Query 7, Line 178 – delete ‘would have’, and line 180, should be ‘did violence increase…’

Reply: Agreed and fixed.

Query 8, Line 182 – should be in relation to (remove would again).

Reply: Agreed and fixed.

Query 9, Line 186 – remove would again… you’re not testing whether anything ‘would’ happen, it’s whether there is evidence that it did happen.

Reply: Agreed and fixed.

Query 10, Line 282 – can you provide a citation for the assertion that butterfly shaped fractures in long-bones can be considered to be perimortem, as well as further clarification justifying this assertion.

Reply: Agreed and fixed. The sentence was deleted, because it is risky to establish a direct relationship between butterfly fractures and perimortem fractures.

Query 11, Line 297 – change wording again from ‘would’, you’re not predicting anything, this is a post-hoc interpretation so a better way to word this is that if you were to see such evidence it ‘could be interpreted as indicative of inter-group violence’.

Reply: Agreed and fixed. The sentence stands now as suggested by the reviewer: The latter scenario could be interpreted as indicative of inter-group violence.

Query 12, Line 349 - Likewise, our investigate (should be investigation).

Reply: Agreed and fixed.

Query 13, Line 577 – I don’t find the argument that unmodified stones in burials may have been weapons to be compelling. Unmodified stones are part of grave fill in just about any burial context anywhere, so this seems to be overly speculative.

Reply: Agreed and fixed. The sentence stands as follows: Around archaic burials, however, it is common to find unmodified spheroidal stones, ranging in size from 3 to 5 cm in diameter (Fig. 7). It is striking that these stones, selected for their shape and size, were placed in the fillings of the mortuary contexts. This means that they were not part of the natural soil matrix, which is fine sand. Consequently, we suggest that these stones could have been thrown and caused the cranial trauma.

Query 14, Line 681 – change devises to devices

Reply: Agreed and fixed.

We hope that, with this new version of the manuscript. we have resolved the critical points raised by the reviewers, to whom we are very grateful for their work.

Best regard

Vivien G. Standen 

corresponding author 

Departamento de Antropología

Universidad de Tarapacá

Arica-Chile

---

## [Decision Letter · Decision Letter 2]

15 Aug 2023

Violence in fishing, hunting, and gathering societies of the Atacama Desert coast: A long-term perspective (10,000 BP - AD 1436).

PONE-D-22-28217R2

Dear Dr. Standen,

We’re pleased to inform you that your manuscript has been judged scientifically suitable for publication and will be formally accepted for publication once it meets all outstanding technical requirements.

Kind regards,

Ron Pinhasi

Academic Editor

PLOS ONE

Additional Editor Comments (optional):

Reviewers' comments:

Reviewer's Responses to Questions

**Comments to the Author**

1. If the authors have adequately addressed your comments raised in a previous round of review and you feel that this manuscript is now acceptable for publication, you may indicate that here to bypass the “Comments to the Author” section, enter your conflict of interest statement in the “Confidential to Editor” section, and submit your "Accept" recommendation.

Reviewer #2: All comments have been addressed

2. Is the manuscript technically sound, and do the data support the conclusions?

Reviewer #2: Yes

3. Has the statistical analysis been performed appropriately and rigorously? 

Reviewer #2: Yes

4. Have the authors made all data underlying the findings in their manuscript fully available?

Reviewer #2: Yes

5. Is the manuscript presented in an intelligible fashion and written in standard English?

Reviewer #2: Yes

6. Review Comments to the Author

Reviewer #2: (No Response)

7. PLOS authors have the option to publish the peer review history of their article (what does this mean?). If published, this will include your full peer review and any attached files.

Reviewer #2: No

---

## [Editor Report · Acceptance letter]

22 Aug 2023

PONE-D-22-28217R2 

Violence in fishing, hunting, and gathering societies of the Atacama Desert coast: A long-term perspective (10,000 BP - AD 1450) 

Dear Dr. Standen:

I'm pleased to inform you that your manuscript has been deemed suitable for publication in PLOS ONE. Congratulations! Your manuscript is now with our production department. 

Kind regards, 

on behalf of

Dr. Ron Pinhasi 

Academic Editor

PLOS ONE